# Revisiting Generalized p-Laplacian Regularized Framelet GCNs: Convergence, Energy Dynamic and as Non-Linear Diffusion

## Abstract

This paper presents a comprehensive theoretical analysis of the graph p-Laplacian regularized framelet network (pL-UFG) to establish a solid understanding of its properties. We conduct a convergence analysis on pL-UFG, addressing the gap in the understanding of its asymptotic behaviors. Further by investigating the generalized Dirichlet energy of pL-UFG, we demonstrate that the Dirichlet energy remains non-zero throughout convergence, ensuring the avoidance of over-smoothing issues. Additionally, we elucidate the energy dynamic perspective, highlighting the synergistic relationship between the implicit layer in pL-UFG and graph framelets. This synergy enhances the model's adaptability to both homophilic and heterophilic data. Notably, we reveal that pL-UFG can be interpreted as a generalized non-linear diffusion process, thereby bridging the gap between pL-UFG and differential equations on the graph. Importantly, these multifaceted analyses lead to unified conclusions that offer novel insights for understanding and implementing pL-UFG, as well as other graph neural network (GNN) models. Finally, based on our dynamic analysis, we propose two novel pL-UFG models with manually controlled energy dynamics. We demonstrate empirically and theoretically that our proposed models not only inherit the advantages of pL-UFG but also significantly reduce computational costs for training on large-scale graph datasets.

## 1 Introduction

Graph neural networks (GNNs) have emerged as a popular tool for the representation learning on the graph-structured data (Wu et al., 2020). To enhance the learning power of GNNs, many attempts have been made by considering the propagation of GNNs via different aspects such as optimization (Zhu et al., 2021; Wei et al., 2022), statistical test (Xu et al., 2019) and gradient flow (Bodnar et al., 2022; Di Giovanni et al., 2022). In particular, treating GNNs propagation as an optimization manner allows one to assign different types of regularizers on the GNNs' output so that the variation of the node features, usually measured by so-called Dirichlet energy, can be properly constrained (Zhu et al., 2021; Chen et al., 2022a). The underlying reason for this regularization operation is due to the recently identified computational issue of GNNs on different types of graphs, namely homophily and heterophily Zheng et al. (2022a). With the former most of the nodes are connected with those nodes with identical labels, and the latter is not Pei et al. (2019). Accordingly, an ideal GNN shall be able to produce a rather smoother node features for homophily graph and more distinguishable node features when the input graph is heterophilic Pei et al. (2019); Bi et al. (2022).

Based on the above statement, a proper design of the regularizer that is flexible to let GNN fit both two types of the graph naturally becomes the next challenge. A recent research Fu et al. (2022b) proposed new energy based regularizer, namely p-Laplacian based regularizer to the optimization of GNN and resulted in an iterative algorithm to approximate the so-called implicit layer induced from the solution of the regularization. To engage a more flexible design of p-Laplacian GNN in (Fu et al., 2022b), Shao et al. (2022) further proposed p-Laplacian based graph framelet GNN (pL-UFG) to assign the p-Laplacian based regularization act on multiscale GNNs (e.g., graph framelet). While remarkable learning accuracy has been observed empirically, the underlying properties of the models proposed in (Shao et al., 2022) are still unclear. In this paper, our

primary focus is on pL-UFG (see Section 2 for the formulation). Our objective is to analyze pL-UFG from various perspectives, including convergence of its implicit layer, model's asymptotic energy behavior, changes of model's dynamics due to the implicit layer, and relationship with existing diffusion models. To the best of our knowledge, these aspects have not been thoroughly explored in the context of p-Laplacian based GNNs, leaving notable knowledge gaps. Accordingly, we summarize our contribution as follows:

- We rigorously prove the convergence of pL-UFG, providing insights into the asymptotic behavior of the model. This analysis addresses a crucial gap in the understanding of GNN models regularized with p-Laplacian based energy regularizer.

- We show that by assigning the proper values of two key model parameters (denoted as $\mu$ and $p$) of pL-UFG based on our theoretical analysis, the (generalized) Dirichlet energy of the node feature produced from pL-UFG will never converge to 0; thus the inclusion of the implicit layer will prevent the model (graph framelet) from potential over-smoothing issue.

- We demonstrate how the implicit layer in pL-UFG interacts with the energy dynamics of the graph framelet. Furthermore, we prove that pL-UFG can adapt to both homophily and heterophily graphs, enhancing its versatility and applicability.

- We establish that the propagation mechanism within pL-UFG enables a generalized non-linear graph diffusion. The conclusions based on our analysis from different perspectives are unified at the end of the paper, suggesting a promising framework for evaluating other GNNs.

- Based on our theoretical results, we propose two generalized pL-UFG models with controlled model dynamics, namely pL-UFG low-frequency dominant model (pL-UFG-LFD) and pL-UFG high frequency dominant model (pL-UFG-HFD). we further show that with controllable model dynamics, the computational cost of pL-UFG is largely reduced, making our proposed model capable of handling large-scale graph datasets.

- We conduct extensive experiments to validate our theoretical claims. The empirical results not only confirm pL-UFG's capability to handle both homophily and heterophily graphs but also demonstrate that our proposed models achieve comparable or superior classification accuracy with reduced computational cost. These findings are consistent across commonly tested and large-scale graph datasets.

The remaining sections of this paper are structured as follows. Section 2 presents fundamental notations related to graphs, GNN models, graph framelets and pL-UFG. In Section 3, we conduct a theoretical analysis on pL-UFG, focusing on the aforementioned aspects. Specifically, Section 3.1 presents the convergence analysis, while Section 3.2 examines the behavior of the p-Laplacian based implicit layer through a generalized Dirichlet energy analysis. Furthermore, Section 3.3 demystifies the interaction between the implicit layer and graph framelets from an energy dynamic perspective. We provide our proposed models (pL-UFG-LFD and pL-UFG-HFD) in section 3.4. Lastly, in Section 3.5, we demonstrate that the iterative algorithm derived from the implicit layer is equivalent to a generalized non-linear diffusion process on the graph. Additionally, in Section 4 we further verify our theoretical claims by comprehensive numerical experiments. Lastly, in conclusion 5, we summarize the findings of this paper and provide suggestions for future research directions.

## 2    Preliminaries

In this section, we provide necessary notations and formulations utilized in this paper. We list the necessary notations with their meanings in the Table 1 below, although we will mention the meaning of them again when we first use them.

Table 1: Necessary notations

| Notations | Brief Interpretation |
| --- | --- |
| $\mathcal{H}(\mathcal{G})$ | Heterophily index of a given graph $\mathcal{G}$ |
| $\mathbf{X}$ | Initial node feature matrix |
| $\mathbf{F}^{(k)}$ | Feature representation on $k$-th layer of GNN model |
| $\mathbf{f}_i$ | Individual row of $\mathbf{F}$ |
| $\mathbf{F}_{i,:}$ | One or more operation acts on each row of $\mathbf{F}$ |
| $\mathbf{D}$ | Graph degree matrix |
| $\widehat{\mathbf{A}}$ | Normalized adjacency matrix |
| $\widetilde{\mathbf{L}}$ | Normalized Laplacian matrix |
| $\mathbf{W}$ | Graph weight matrix |
| $\mathcal{W}$ | Framelet decomposition matrix |
| $\mathcal{I}$ | Index set of all framelet decomposition matrices. |
| $\widehat{\mathbf{W}}$ | Learnable weight matrix in GNN models |
| $\widetilde{\mathbf{W}},\mathbf{\Omega},\widehat{\mathbf{W}}$ | Learnable weight matrices in defining generalized Dirichlet energy |
| $\mathbf{Y}$ | Feature propagation result for the pL-UFG defined in Shao et al. (2022). |
| $\theta$ | N-dimensional vector for diagonal scaling $(\mathrm{diag}(\theta))$ in framelet models. |
| $\mathbf{E}^{PF}(\mathbf{F})$ | Generalized Dirichlet energy for node feature induced from implicit layer |
| $\mathbf{E}^{Fr}(\mathbf{F})$ | Generalized framelet Dirichlet energy |
| $\mathbf{E}^{total}(\mathbf{F})$ | Total generalized Dirichlet energy |
| $\{\lambda_i, \mathbf{u}_i\}_{i=1}^N$ | Eigen-pairs of $\widetilde{\mathbf{L}}$ |

We also provide essential background information on the developmental history before the formulation of certain models, serving as a concise introduction to the related works.

**Graph, Graph Convolution and Graph Consistency**  We denote a weighted graph as $\mathcal{G} = (\mathcal{V}, \mathcal{E}, \mathbf{W})$ with nodes set $\mathcal{V} = \{v_1, v_2, \cdots, v_N\}$ of total $N$ nodes, edge set $\mathcal{E} \subseteq \mathcal{V} \times \mathcal{V}$ and graph adjacency matrix $\mathbf{W}$, where $\mathbf{W} = [w_{i,j}] \in \mathbb{R}^{N \times N}$ and $w_{i,j} = 1$ if $(v_i, v_j) \in \mathcal{E}$, else, $w_{i,j} = 0$. The nodes feature matrix is $\mathbf{X} \in \mathbb{R}^{N \times c}$ for $\mathcal{G}$ with each row $\mathbf{x}_i \in \mathbb{R}^c$ as the feature vector associated with node $v_i$. For a matrix $\mathbf{A}$, we denote its transpose as $\mathbf{A}^\top$, and we use $[N]$ for set $\{1, 2, \ldots, N\}$. Throughout this paper, we will only focus on the undirect graph and use matrix $\mathbf{A}$ and $\mathbf{W}$ interchangeably for graph adjacency matrix[1]. The normalized graph Laplacian is defined as $\widetilde{\mathbf{L}} = \mathbf{I} - \mathbf{D}^{-\frac{1}{2}}(\mathbf{W} + \mathbf{I})\mathbf{D}^{-\frac{1}{2}}$, where $\mathbf{D} = \mathrm{diag}(d_{1,1}, \ldots, d_{N,N})$ is a diagonal degree matrix with $d_{i,i} = \sum_{j=1}^N w_{i,j}$ for $i = 1, \ldots, N$, and $\mathbf{I}$ is the identity matrix. From the spectral graph theory (Chung, 1997), we have $\widetilde{\mathbf{L}} \succeq 0$, i.e. $\widetilde{\mathbf{L}}$ is a positive semi-definite (SPD) matrix. Let $\{\lambda_i\}_{i=1}^N$ in decreasing order be all the eigenvalues of $\widetilde{\mathbf{L}}$, also known as graph spectra, and $\lambda_i \in [0, 2]$. For any given graph, we let $\rho_{\widetilde{\mathbf{L}}}$ be the largest eigenvalue of $\widetilde{\mathbf{L}}$. Lastly, for any vector $\mathbf{x} = [x_1, ..., x_c] \in \mathbb{R}^c$, $\|\mathbf{x}\|_2 = (\sum_{i=1}^c x_i^2)^{\frac{1}{2}}$ is the L$_2$-norm of $\mathbf{x}$, and similarly, for any matrix $\mathbf{M} = [m_{i,j}]$, denote by $\|\mathbf{M}\| := \|\mathbf{M}\|_F = (\sum_{i,j} m_{i,j}^2)^{\frac{1}{2}}$ the matrix Frobenius norm.

Graph convolution network (GCN) (Kipf & Welling, 2016) produces a layer-wise (node feature) propagation rule based on the information from the normalized adjacency matrix as:

$$\mathbf{F}^{(k+1)} = \sigma\big(\widehat{\mathbf{A}}\mathbf{F}^{(k)}\widehat{\mathbf{W}}^{(k)}\big), \tag{1}$$

where $\mathbf{F}^{(k)}$ is the embedded node feature, $\widehat{\mathbf{W}}^{(k)}$ the weight matrix for channel mixing (Bronstein et al., 2021), and $\sigma$ any activation function such as sigmoid. The superscript $^{(k)}$ indicates the quantity associated with layer $k$, and $\mathbf{F}^{(0)} = \mathbf{X}$. We write $\widehat{\mathbf{A}} = \mathbf{D}^{-\frac{1}{2}}(\mathbf{W} + \mathbf{I})\mathbf{D}^{-\frac{1}{2}}$, the normalized adjacency matrix of $\mathcal{G}$. It is easy to see that the operation conducted in GCN before activation can be interpreted as a localized filter by the graph Fourier transform, i.e., $\mathbf{F}^{(k+1)} = \mathbf{U}(\mathbf{I}_n - \mathbf{\Lambda})\mathbf{U}^\top \mathbf{F}^{(k)}$, where $\mathbf{U}, \mathbf{\Lambda}$ are from the eigendecomposition $\widetilde{\mathbf{L}} = \mathbf{U}\mathbf{\Lambda}\mathbf{U}^\top$. In fact, $\mathbf{U}\mathbf{F}$ is known as the Fourier transform of graph signals in $\mathbf{F}$.

---

[1]We initially set $\mathbf{W}$ as the graph adjacency matrix while $\mathbf{W}$ is a generic edge weight matrix in align with the notations used in (Fu et al., 2022a; Shao et al., 2022)

Over the development of GNNs, most of GNNs are designed under the homophily assumption in which connected (neighbouring) nodes are more likely to share the same label. The recent work by Zhu et al. (2020) identifies that the general topology GNN fails to obtain outstanding results on the graphs with different class labels and dissimilar features in their connected nodes, such as the so-call heterophilic graphs. The definition of homophilic and heterophilic graphs are given by:

**Definition 1** (Homophily and Heterophily (Fu et al., 2022a)). *The homophily or heterophily of a network is used to define the relationship between labels of connected nodes. The level of homophily of a graph can be measured by $\mathcal{H}(\mathcal{G}) = \mathbb{E}_{v_i \in \mathcal{V}}[|\{v_j\}_{j \in \mathcal{N}_i, y_i = y_i}|/|\mathcal{N}_i|]$, where $|\{v_j\}_{j \in \mathcal{N}_i, y_i = y_i}|$ denotes the number of neighbours of $v_i \in \mathcal{V}$ that share the same label as $v_i$, i.e. $y_i = y_j$. $\mathcal{H}(\mathcal{G}) \to 1$ corresponds to strong homophily while $\mathcal{H}(\mathcal{G}) \to 0$ indicates strong heterophily. We say that a graph is a homophilic (heterophilic) graph if it has strong homophily (heterophily).*

**Graph Framelet.** As the main target for this paper to explore is pL-UFG defined in (Shao et al., 2022) in which p-Laplacian based implicit layer is combined with so-called graph framelet or framelets in short. Framelets are a type of wavelet frames arising from signal processing which can be extended for analysing graph signals. The first wavelet frame with a lifting scheme for graph analysis was presented in (Sweldens, 1998). As computational power increased, Hammond et al. (2011) proposed a framework for wavelet transformation on graphs using Chebyshev polynomials for approximations. Later, Dong (2017) developed tight framelets on graphs by approximating smooth functions with filtered Chebyshev polynomials.

Framelets have been applied to graph learning tasks with outstanding results, as demonstrated in (Zheng et al., 2021). They are capable of decomposing graph signals and re-aggregating them effectively, as shown in the study on graph noise reduction by Zhou et al. (2021) Combining framelets with singular value decomposition (SVD) has also made them applicable to directed graphs (Zou et al., 2022). Recently, Yang et al. (2022) suggested a simple method for building more versatile and stable framelet families, known as Quasi-Framelets. In this study, we will introduce graph framelets using the same architecture described in (Yang et al., 2022). To begin, we define the filtering functions for Quasi-framelets.

**Definition 2.** *A set of $R + 1$ positive functions $\mathcal{F} = \{g_0(\xi), g_1(\xi), ..., g_R(\xi)\}$ defined on the interval $[0, \pi]$ is considered as (a set of) Quasi-Framelet scaling functions, if these functions adhere to the following identity condition:*

$$g_0(\xi)^2 + g_1(\xi)^2 + \cdots + g_R(\xi)^2 \equiv 1, \quad \forall \xi \in [0, \pi]. \tag{2}$$

The identity condition eq. (2) ensures a perfect reconstruction of a signal from its spectral space to the spatial space, see (Yang et al., 2022) for a proof. Particularly we are interested in the scaling function set in which $g_0$ descents from 1 to 0, i.e., $g_0(0) = 1$ and $g_0(\pi) = 0$ and $g_R$ ascends from 0 to 1, i.e., $g_R(0) = 0$ and $g_R(\pi) = 1$. The purpose of setting these conditions is for $g_0$ to regulate the highest frequency and for $g_R$ to control the lowest frequency, while the remaining functions govern the frequencies lying between them.

With a given set of framelet scaling functions, the so-called Quasi-Framelet signal transformation can be defined by the following transformation matrices:

$$\mathcal{W}_{0,J} = \mathbf{U} g_0(\frac{\mathbf{\Lambda}}{2^{m+J}}) \cdots g_0(\frac{\mathbf{\Lambda}}{2^m}) \mathbf{U}^\top, \tag{3}$$

$$\mathcal{W}_{r,0} = \mathbf{U} g_r(\frac{\mathbf{\Lambda}}{2^m}) \mathbf{U}^\top, \quad \text{for } r = 1, ..., R, \tag{4}$$

$$\mathcal{W}_{r,\ell} = \mathbf{U} g_r(\frac{\mathbf{\Lambda}}{2^{m+\ell}}) g_0(\frac{\mathbf{\Lambda}}{2^{m+\ell-1}}) \cdots g_0(\frac{\mathbf{\Lambda}}{2^m}) \mathbf{U}^\top, \tag{5}$$

$$\text{for } r = 1, ..., R, \ell = 1, ..., J,$$

where $\mathcal{F}$ is a given set of Quasi-Framelet functions satisfying eq. (2) and $J \geq 0$ is a given level on a graph $\mathcal{G} = (\mathcal{V}, \mathcal{E})$ with normalized graph Laplacian $\widetilde{\mathbf{L}} = \mathbf{U}^\top \mathbf{\Lambda} \mathbf{U}$. $\mathcal{W}_{0,J}$ is defined as the product of $J + 1$ Quasi-Framelet scaling functions $g_0$ applied to the Laplacian spectra $\mathbf{\Lambda}$ at different scales. defined as $g_r(\frac{\mathbf{\Lambda}}{2^m})$ applied to spectra $\mathbf{\Lambda}$, where $m$ is the coarsest scale level which is the smallest value satisfying $2^{-m} \lambda_n \leq \pi$. For $1 \leq r \leq R$ and $1 \leq \ell \leq J$, $\mathcal{W}_{r,\ell}$ is defined as the product of $J - \ell + 1$ Quasi-Framelet scaling functions $g_0$ and $\ell$ Quasi-Framelet scaling functions $g_r$ applied to spectra $\mathbf{\Lambda}$.

Let $\mathcal{W} = [\mathcal{W}_{0,J}; \mathcal{W}_{1,0}; ...; \mathcal{W}_{R,0}]$ be the stacked matrix. It can be proven that $\mathcal{W}^T\mathcal{W} = \mathbf{I}$, see (Yang et al., 2022), which provides a signal decomposition and reconstruction process based on $\mathcal{W}$. This is referred to as the graph Quasi-Framelet transformation.

Since the computation of the Quasi-framelet transformation matrices requires the eigendecomposition of graph Laplacian, to reduce the computational cost, Chebyshev polynomials are used to approximate the Quasi-Framelet transformation matrices. The approximated transformation matrices are defined by replacing $g_r(\xi)$ in eq. (3)-eq. (5) with Chebyshev polynomials $\mathcal{T}_r(\xi)$ of a fixed degree, which is typically set to 3. The Quasi-Framelet transformation matrices defined in eq. (3) - eq. (5) can be approximated by,

$$\mathcal{W}_{0,J} \approx \mathcal{T}_0(\frac{1}{2^{m+J}}\widetilde{\mathbf{L}}) \cdots \mathcal{T}_0(\frac{1}{2^m}\widetilde{\mathbf{L}}), \tag{6}$$

$$\mathcal{W}_{r,0} \approx \mathcal{T}_r(\frac{1}{2^m}\widetilde{\mathbf{L}}), \quad \text{for } r = 1, ..., R, \tag{7}$$

$$\mathcal{W}_{r,\ell} \approx \mathcal{T}_r(\frac{1}{2^{m+\ell}}\widetilde{\mathbf{L}})\mathcal{T}_0(\frac{1}{2^{m+\ell-1}}\widetilde{\mathbf{L}}) \cdots \mathcal{T}_0(\frac{1}{2^m}\widetilde{\mathbf{L}}), \tag{8}$$

$$\text{for } r = 1, ..., R, \ell = 1, ..., J.$$

Based on the approximated Quasi-Framelet transformation defined above, two types of graph framelet convolutions have been developed recently:

1. **The Spectral Framelet Models** (Zheng et al., 2021; 2022b; Yang et al., 2022; Shi et al., 2023):

$$\mathbf{F}^{(k+1)} = \sigma\left(\mathcal{W}^\top \text{diag}(\theta)\mathcal{W}\mathbf{F}^{(k)}\right) := \sigma\left(\sum_{(r,\ell)\in\mathcal{I}} \mathcal{W}_{r,\ell}^\top \text{diag}(\theta_{r,\ell})\mathcal{W}_{r,\ell}\mathbf{F}^{(k)}\widehat{\mathbf{W}}^{(k)}\right), \tag{9}$$

where $\theta_{r,\ell} \in \mathbb{R}^N$, $\widehat{\mathbf{W}}^{(k)}$ are learnable matrices for channel/feature mixing, and $\mathcal{I} = \{(r,j) : r = 1, ..., R, \ell = 0, 1, ..., J\} \cup \{(0, J)\}$ is the index set for all framelet decomposition matrices.

2. **The Spatial Framelet Models** (Chen et al., 2022a):

$$\mathbf{F}^{(k+1)} = \sigma\left(\mathcal{W}_{0,J}^\top\widehat{\mathbf{A}}\mathcal{W}_{0,J}\mathbf{F}^{(k)}\widehat{\mathbf{W}}_{0,J}^{(k)} + \sum_{r,\ell} \mathcal{W}_{r,\ell}^\top\widehat{\mathbf{A}}\mathcal{W}_{r,\ell}\mathbf{F}^{(k)}\widehat{\mathbf{W}}_{r,\ell}^{(k)}\right). \tag{10}$$

The spectral framelet models conduct framelet decomposition and reconstruction on the spectral domain of the graph. Clearly $\theta_{r,\ell} \in \mathbb{R}^N$ can be interpreted as the frequency filters, given that the framelet system provides a perfect reconstruction on the input graph signal (i.e., $\mathcal{W}^\top\mathcal{W} = \mathbf{I}$). Instead of frequency domain filtering, the spatial framelet models implement the framelet-based propagation via spatial (graph adjacency) domain.

There is a major difference between two schemes. In the spectral framelet methods, the weight matrix $\widehat{\mathbf{W}}^{(k)}$ is shared across different (filtered) frequency domains, while in the spatial framelet methods, an individual weight matrix $\widehat{\mathbf{W}}_{r,\ell}^{(k)}$ is applied to each (filtered) spatial domain to produce the graph convolution.

Finally, it is worth to noting that applying framelet/quasi-framelet transforms on graph signals can decomposes graph signals on different frequency domains for processing, e.g., the filtering used in the spectral framelet models and the spatial aggregating used in the spatial framelet models, thus the perfect reconstruction property guarantees less information loss in the signal processing pipeline. The learning advantage of graph framelet models has been proved via both theoretical and empirical studies (Han et al., 2022; Zheng et al., 2021; Chen et al., 2022a).

**Generalized p-Laplacian Regularized Framelet GCN.** In this part, we provide several additional definitions to formulate the model (pL-UFG) that we are interested in analyzing. As a generalized framelet model incorporating with the so-called $p$-Laplacian energy regularizer, the pL-UFG, as we are going to

define it later, has shown great flexibility in terms of adapting different types of graphs (i.e., homophily and heterophily) by efficiently adjusting the penalty strength from the regularizer, resulted in superior learning performance across various benchmark datasets (Shao et al., 2022). To thoroughly define pL-UFG, we start by defining the $p$-Laplace operator as follows.

**Definition 3** (The $p$-Laplace Operator (Drábek & Pohozaev, 1997)). *Let $\Omega \subset \mathbb{R}^d$ be a domain and $u$ is a function defined on $\Omega$. The p-Laplace operator $\Delta$ over functions is defined as*

$$\Delta u := \nabla \cdot (\|\nabla u\|^{p-2} \nabla u)$$

*where $\nabla$ is the gradient operator and $\|\cdot\|$ is the Euclidean norm and $p$ is a scalar satisfying $1 < p < +\infty$. The p-Laplace operator, is known as a quasi-linear elliptic partial differential operator.*

There are a line of research on the properties of $p$-Laplacian in regarding to its uniqueness and existence (García Azorero & Peral Alonso, 1987), geometrical property (Kawohl & Horak, 2016) and boundary conditions on so-called p-Laplacian equation (Torres, 2014).

The concept of $p$-Laplace operator can be extended for discrete domains such as graph (nodes) based on the concepts of the so-called graph gradient and divergence, see below, one of the recent works Fu et al. (2022a) considers assigning an adjustable $p$-Laplacian regularizer to the (discrete) graph regularization problem that is conventionally treated as a way of producing GNN outcomes (i.e., Laplacian smoothing) (Zhou & Schölkopf, 2005). In view of the fact that the classic graph Laplacian regularizer measures the graph signal energy along edges under $L_2$ metric, it would be beneficial if GNN training process can be regularized under $L_p$ metric in order to adapt to different graph inputs. Following these pioneer works, Shao et al. (2022) further integrated graph framelet and a generalized $p$-Laplacian regularizer to develop the so-called generalized $p$-Laplacian regularized framelet model. It involves a regularization problem over the energy quadratic form induced from the graph $p$-Laplacian. To show this, we start by defining graph gradient as follows:

To introduce graph gradient and divergence, we define the following notation:

Given a graph $\mathcal{G} = (\mathcal{V}, \mathcal{E}, \mathbf{W})$, let $\mathcal{F}_\mathcal{V} := \{\mathbf{F}|\mathbf{F} : \mathcal{V} \to \mathbb{R}^d\}$ be the space of the vector-valued functions defined on $\mathcal{V}$ and $\mathcal{F}_\mathcal{E} := \{\mathbf{g}|\mathbf{g} : \mathcal{E} \to \mathbb{R}^d\}$ be the vector-valued function space on edges, respectively.

**Definition 4** (Graph Gradient (Zhou & Schölkopf, 2005)). *For a given function $\mathbf{F} \in \mathcal{F}_\mathcal{V}$, its graph gradient is an operator $\nabla_W : \mathcal{F}_\mathcal{V} \to \mathcal{F}_\mathcal{E}$ defined as for all $(v_i, v_j) \in \mathcal{E}$,*

$$(\nabla_W \mathbf{F})([i,j]) := \sqrt{\frac{w_{i,j}}{d_{j,j}}} \mathbf{f}_j - \sqrt{\frac{w_{i,j}}{d_{i,i}}} \mathbf{f}_i, \tag{11}$$

*where $\mathbf{f}_i$ and $\mathbf{f}_j$ are the signal vectors on nodes $v_i$ and $v_j$, i.e., the rows of $\mathbf{F}$.*

For simplicity, we denote $\nabla_W \mathbf{F}$ as $\nabla \mathbf{F}$ as the graph gradient. The definition of (discrete) graph gradient is analogized from the notion of gradient from the continuous space. Similarly, we can further define the so-called graph divergence:

**Definition 5** (Graph Divergence (Zhou & Schölkopf, 2005)). *The graph divergence is an operator $div : \mathcal{F}_\mathcal{E} \to \mathcal{F}_\mathcal{V}$ which is defined by the following way. For a given function $\mathbf{g} \in \mathcal{F}_\mathcal{E}$, the resulting $div(\mathbf{g}) \in \mathcal{F}_\mathcal{V}$ satisfies the following condition, for any functions $\mathbf{F} \in \mathcal{F}_\mathcal{V}$,*

$$\langle \nabla \mathbf{F}, \mathbf{g} \rangle = \langle \mathbf{F}, -div(\mathbf{g}) \rangle. \tag{12}$$

It is easy to check that the graph divergence can be computed by:

$$\text{div}(\mathbf{g})(i) = \sum_{j=1}^{N} \sqrt{\frac{w_{i,j}}{d_{i,i}}} (\mathbf{g}[i,j] - \mathbf{g}[j,i]). \tag{13}$$

With the formulation of graph gradient and divergence we are ready to define the graph p-Laplacian operator and the corresponding p-Dirichlet form (Zhou & Schölkopf, 2005; Fu et al., 2022a) that serves as the regularizer in the model developed in (Shao et al., 2022). The graph p-Laplacian can be defined as follows:

**Definition 6** (Graph $p$-Laplacian). *Given a graph $\mathcal{G} = (\mathcal{V}, \mathcal{E}, \mathbf{W})$ and a multiple channel signal function $\mathbf{F} : \mathcal{V} \to \mathbb{R}^d$, the graph $p$-Laplacian is an operator $\Delta_p : \mathcal{F}_{\mathcal{V}} \to \mathcal{F}_{\mathcal{V}}$, defined by:*

$$\Delta_p \mathbf{F} := -\frac{1}{2} div(\|\nabla \mathbf{F}\|^{p-2} \nabla \mathbf{F}), \quad for \ p \geq 1. \tag{14}$$

*where $\| \cdot \|^{p-2}$ is element-wise power over the node gradient $\nabla \mathbf{F}$.*

The corresponding p-Dirichlet form can be denoted as:

$$\mathcal{S}_p(\mathbf{F}) = \frac{1}{2} \sum_{(v_i, v_j) \in \mathcal{E}} \left\| \sqrt{\frac{w_{i,j}}{d_{j,j}}} \mathbf{f}_j - \sqrt{\frac{w_{i,j}}{d_{i,i}}} \mathbf{f}_i \right\|^p, \tag{15}$$

where we adopt the definition of $p$-norm as (Fu et al., 2022a). It is not difficult to verify that once we set $p = 2$, we recover the graph Dirichlet energy (Zhou & Schölkopf, 2005) that is widely used to measure the difference between node features along the GNN propagation process.

**Remark 1** (Dirichlet Energy, Graph Homophily and Heterophily). Graph Dirichlet energy (Fu et al., 2022a; Bronstein et al., 2021) has become a commonly applied measure of variation between node features via GNNs. It has been shown that once the graph is highly heterophilic where the connected nodes are not likely to share identical labels, one may prefer GNNs that exhibit nodes feature sharpening effect, thus increasing Dirichlet energy, such that the final classification output of the connected nodes from these GNNs tend to be different. Whereas, when the graph is highly homophilic, a smoothing effect (thus a decrease of Dirichlet energy) is preferred.

Shao et al. (2022) further generalized the p-Dirichlet form in eq. (15) as:

$$\mathcal{S}_p(\mathbf{F}) = \frac{1}{2} \sum_{(v_i, v_j) \in \mathcal{E}} \|\nabla_W \mathbf{F}([i,j])\|^p$$

$$= \frac{1}{2} \sum_{v_i \in \mathcal{V}} \left[ \left( \sum_{v_j \sim v_i} \|\nabla_W \mathbf{F}([i,j])\|^p \right)^{\frac{1}{p}} \right]^p = \frac{1}{2} \sum_{v_i \in \mathcal{V}} \|\nabla_W \mathbf{F}(v_i)\|_p^p, \tag{16}$$

where $v_j \sim v_i$ stands for the node $v_j$ that is connected to node $v_i$ and $\nabla_W \mathbf{F}(v_i) = (\nabla_W \mathbf{F}([i,j]))_{v_j : (v_i, v_j) \in \mathcal{E}}$ is the node gradient vector for each node $v_i$ and $\| \cdot \|_p$ is the vector $p$-norm. Moreover, we can further generalize the regularizer $\mathcal{S}_p(\mathbf{F})$ by considering any positive convex function $\phi$ as:

$$\mathcal{S}_p^\phi(\mathbf{F}) = \frac{1}{2} \sum_{v_i \in \mathcal{V}} \phi(\|\nabla_W \mathbf{F}(v_i)\|_p). \tag{17}$$

There are many choices of $\phi$ and $p$. When $\phi(\xi) = \xi^p$, we recover the $p$-Laplacian regularizer. Interestingly, by setting $\phi(\xi) = \xi^2$, we recover the so-called Tikhonov regularization which is frequently applied in image processing. When $\phi(\xi) = \xi$, i.e. identity map written as $id$, and $p = 1$, $\mathcal{S}_1^{id}(\mathbf{F})$ becomes the classic total variation regularization. Last but not the least, $\phi(\xi) = r^2 \log(1 + \xi^2/r^2)$ gives nonlinear diffusion. We note that there are many other choices on the form of $\phi$. In this paper we will only focus on those mentioned in Shao et al. (2022) (i.e., the smooth ones). As a result, the flexible design of the **p-Laplacian based energy regularizer** in eq. (17) provides different penalty strength in regularizing the node features propagated from GNNs.

Accordingly, the regularization problem proposed in Shao et al. (2022) is:

$$\mathbf{F} = \underset{\mathbf{F}}{\arg\min} \ \mathcal{S}_p^\phi(\mathbf{F}) + \mu \|\mathbf{F} - \mathcal{W}^\top \text{diag}(\theta) \mathcal{W} \mathbf{F}^{(k)}\|_F^2, \tag{18}$$

where $\mathbf{Y} := \mathcal{W}^\top \text{diag}(\theta) \mathcal{W} \mathbf{F}^{(k)}$ stands for the node feature generated by the spectral framelet models (9) without activation $\sigma$. This is the implicit layer proposed in Shao et al. (2022). As the optimization problem defined in eq. (18) does not have a closed-form solution when $p \neq 2$, an iterative algorithm is developed in Shao et al. (2022) to address this issue. The justification is summarized by the following proposition (Theorem 1 in Shao et al. (2022)):

**Proposition 1.** *For a given positive convex function $\phi(\xi)$, define*

$$M_{i,j} = \frac{w_{i,j}}{2} \left\| \nabla_W \mathbf{F}([i,j]) \right\|^{p-2} \cdot \left[ \frac{\phi'(\|\nabla_W \mathbf{F}(v_i)\|_p)}{\|\nabla_W \mathbf{F}(v_i)\|_p^{p-1}} + \frac{\phi'(\|\nabla_W \mathbf{F}(v_j)\|_p)}{\|\nabla_W \mathbf{F}(v_j)\|_p^{p-1}} \right],$$

$$\alpha_{ii} = 1 / \left( \sum_{v_j \sim v_i} \frac{M_{i,j}}{d_{i,i}} + 2\mu \right), \qquad \beta_{ii} = 2\mu\alpha_{ii},$$

*and denote the matrices $\mathbf{M} = [M_{i,j}]$, $\boldsymbol{\alpha} = diag(\alpha_{11}, ..., \alpha_{NN})$ and $\boldsymbol{\beta} = diag(\beta_{11}, ..., \beta_{NN})$. Then problem (18) can be solved by the following message passing process*

$$\mathbf{F}^{(k+1)} = \boldsymbol{\alpha}^{(k)} \mathbf{D}^{-1/2} \mathbf{M}^{(k)} \mathbf{D}^{-1/2} \mathbf{F}^{(k)} + \boldsymbol{\beta}^{(k)} \mathbf{Y}, \tag{19}$$

*with an initial value, e.g., $\mathbf{F}^{(0)} = \mathbf{0}$ or $\mathbf{Y}$. Note, k denotes the discrete time index (iteration).*

Finally, in this paper, we call the the iterative algorithm defined in eq. (19) for realizing the implicit layer of the problem defined in eq. (18) as the **pL-UFG** model. Although remarkable performance have been observed from pL-UFG, there are still some key properties of the model that require to be explicitly presented. For example, the convergence of the iterative algorithm eq. (19) for the implicit layer in eq. (18), and how the implicit layer changes and interacts with the energy dynamic of the original framelet, what is the relationship between the propagation within implicit layer and other propagations such as diffusion on node features. We will explicitly show our theoretical results in the coming sections.

## 3 Theoretical Analysis of the pL-UFG

In this section, we show detailed analysis on the convergence (Section 3.1) and energy behavior (Section 3.2) of the iterative algorithm in solving the implicit layer presented in eq. (19). In addition, we will also present some results regarding to the interaction between the implicit layer and graph framelet in Section 3.3 via the energy dynamic aspect based on the conclusion from Section 3.2. Lastly in Section 3.5, we will verify that the iterative algorithm induced from the p-Laplacian implicit layer admits a generalized non-linear diffusion process, thereby connecting the discrete iterative algorithm to the differential equations on graph.

First, we consider the form of matrix $\mathbf{M}$ in eq. (19). Write

$$\zeta_{i,j}^{\phi}(\mathbf{F}) = \frac{1}{2} \left[ \frac{\phi'(\|\nabla_W \mathbf{F}^{(k+1)}(v_i)\|_p)}{\|\nabla_W \mathbf{F}^{(k+1)}(v_i)\|_p^{p-1}} + \frac{\phi'(\|\nabla_W \mathbf{F}^{(k+1)}(v_j)\|_p)}{\|\nabla_W \mathbf{F}^{(k+1)}(v_j)\|_p^{p-1}} \right]. \tag{20}$$

$M_{i,j}$ can be simplified as

$$M_{i,j} = \zeta_{i,j}^{\phi}(\mathbf{F}) w_{i,j} \left\| \nabla_W \mathbf{F}([i,j]) \right\|^{p-2}. \tag{21}$$

$\zeta_{i,j}^{\phi}(\mathbf{F})$ is bounded as shown in the following lemma.

**Lemma 1.** *Assume*

$$\frac{\phi'(\xi)}{\xi^{p-1}} \le C, \tag{22}$$

*for a suitable constant C. We have $|\zeta_{i,j}^{\phi}(\mathbf{F})| \le C$.*

The proof is trivial thus we omit it here. In the sequel, we use $\zeta_{i,j}(\mathbf{F})$ for $\zeta_{i,j}^{\phi}(\mathbf{F})$ instead.

**Remark 2.** It is reasonable for assuming condition in (22) in Lemma 1 so that $\zeta_{i,j}(\mathbf{F})$ is bounded. For example, one can easily verify that when $\phi(\xi) = \xi^p$, $\zeta_{i,j}(\mathbf{F})$ is bounded for all $p$. In particular, when $p = 2$, i.e., $\phi(\xi) = \xi^2$, we have $\frac{\phi'(\xi)}{\xi^{p-1}} = \frac{2\xi}{\xi^{p-1}} = 2\frac{\xi^2}{\xi^p}$, thus $\zeta_{i,j}(\mathbf{F})$ is bounded for all $0 < p \le 2$. Furthermore, when $\phi(\xi) = \xi$, then $\frac{\phi'(\xi)}{\xi^{p-1}} = \frac{\xi}{\xi^{p-1}}$, indicating $\zeta_{i,j}(\mathbf{F})$ is bounded for all $0 < p \le 1$. In addition, when

$\phi(\xi) = \sqrt{\xi^2 + \epsilon^2} - \epsilon$, we have $\frac{\phi'(\xi)}{\xi^{p-1}} = \frac{(\xi^2+\epsilon^2)^{1/2}\xi}{\xi^{p-1}} \leq C\frac{\xi}{\xi^{p-1}}$. Therefore $\zeta_{i,j}(\mathbf{F})$ is bounded for all $0 < p \leq 2$. Lastly, when $\phi(\xi) = r^2 \log(1 + \frac{\xi^2}{r^2})$, the result of $\frac{\phi'(\xi)}{\xi^{p-1}}$ yields $r^2\frac{1}{1+\frac{\xi^2}{r^2}} \cdot \frac{\frac{2}{r^2}\xi}{\xi^{p-1}} \leq 2\frac{\xi}{\xi^{p-1}}$. Hence $\zeta_{i,j}(\mathbf{F})$ remain bounded for all $0 < p \leq 2$. In summary, for all forms of $\phi$ we included in the model, $\zeta_{i,j}(\mathbf{F})$ is bounded.

The boundedness of $\zeta_{i,j}(\mathbf{F})$ from Lemma 1 is useful in the following convergence analysis.

### 3.1 Convergence Analysis of pL-UFG

We show the iterative algorithm presented in eq. (19) will converge with a suitable choice of $\mu$. We further note that although the format of Theorem 1 is similar to Theorem 2 in (Fu et al., 2022b), our message passing scheme presented in eq. (19) is different compared to the one defined in (Fu et al., 2022b) via the forms of $\mathbf{M}$, $\boldsymbol{\alpha}$ and $\boldsymbol{\beta}$. In fact, the model defined in (Fu et al., 2022b) can be considered as a special case where $\phi(\xi) = \xi^p$. As a generalization of the model proposed in (Fu et al., 2022b), we provide a uniform convergence analysis for the pL-UFG.

**Theorem 1** (Weak Convergence of the Proposed Model). *Given a graph $\mathcal{G}(\mathcal{V}, \mathcal{E}, \mathbf{W})$ with node features $\mathbf{X}$, if $\boldsymbol{\alpha}^{(k)}$, $\boldsymbol{\beta}^{(k)}$, $\mathbf{M}^{(k)}$ and $\mathbf{F}^{(k)}$ are updated according to eq. (19), then there exist some real positive value $\mu$, which depends on the input graph $(\mathcal{G}, \mathbf{X})$ and the quantity of p, updated in each iteration, such that:*

$$\mathcal{L}_p^\phi(\mathbf{F}^{(k+1)}) \leq \mathcal{L}_p^\phi(\mathbf{F}^{(k)}),$$

*where $\mathcal{L}_p^\phi(\mathbf{F}) := \mathcal{S}_p^\phi(\mathbf{F}) + \mu\|\mathbf{F} - \mathbf{Y}\|_F^2$.*

*Proof.* First, write

$$M_{i,j}^{(k)} = \frac{w_{i,j}}{2} \left\| \nabla_W \mathbf{F}^{(k)}([i,j]) \right\|^{p-2} \cdot \left[ \frac{\phi'(\|\nabla_W \mathbf{F}^{(k)}(v_i)\|_p)}{\|\nabla_W \mathbf{F}^{(k)}(v_i)\|_p^{p-1}} + \frac{\phi'(\|\nabla_W \mathbf{F}^{(k)}(v_j)\|_p)}{\|\nabla_W \mathbf{F}^{(k)}(v_j)\|_p^{p-1}} \right]. \tag{23}$$

The derivative of the regularization problem defined in eq. (18) is:

$$
\begin{aligned}
\left.\frac{\partial \mathcal{L}_p^\phi(\mathbf{F})}{\partial \mathbf{F}_{i,:}}\right|_{\mathbf{F}^{(k)}} &= 2\mu(\mathbf{F}_{i,:}^{(k)} - \mathbf{Y}_{i,:}) + \sum_{v_j \sim v_i} M_{ij}^{(k)} \frac{1}{\sqrt{d_{ii}w_{ij}}} \nabla_W \mathbf{F}^{(k)}([j,i]) \\
&= 2\mu(\mathbf{F}_{i,:}^{(k)} - \mathbf{Y}_{i,:}) + \sum_{v_j \sim v_i} M_{ij}^{(k)} \left( \frac{1}{d_{ii}}\mathbf{F}_{i,:}^{(k)} - \frac{1}{\sqrt{d_{ii}d_{jj}}}\mathbf{F}_{j,:}^{(k)} \right) \\
&= (2\mu + \sum_{v_j \sim v_i} M_{ij}^{(k)}/d_{ii})\mathbf{F}_{i,:}^{(k)} - 2\mu\mathbf{Y}_{i,:} - \sum_{v_j \sim v_i} \frac{M_{ij}^{(k)}}{\sqrt{d_{ii}d_{jj}}}\mathbf{F}_{j,:}^{(k)} \\
&= \frac{1}{\alpha_{ii}^{(k)}}\mathbf{F}_{i,:}^{(k)} - \frac{1}{\alpha_{ii}^{(k)}} \left( \beta_{ii}^{(k)}\mathbf{Y}_{i,:} + \alpha_{ii}^{(k)} \sum_{v_j \sim v_i} \frac{M_{ij}^{(k)}}{\sqrt{d_{ii}d_{jj}}}\mathbf{F}_{j,:}^{(k)} \right)
\end{aligned}
\tag{24}
$$

Thus, according the update rule of $\mathbf{F}^{(k+1)}$ in eq. (19), we have

$$\left.\frac{\partial \mathcal{L}_p^\phi(\mathbf{F})}{\partial \mathbf{F}_{i,:}}\right|_{\mathbf{F}^{(k)}} = \frac{\mathbf{F}_{i,:}^{(k)} - \mathbf{F}_{i,:}^{(k+1)}}{\alpha_{ii}^{(k)}}. \tag{25}$$

For our purpose, we denote the partial derivative at $\mathbf{F}^{(*)}$ of the objective function with respect to the node feature $\mathbf{F}_{i,;}$ as

$$\partial \mathcal{L}_p^\phi(\mathbf{F}_{i,:}^{(*)}) := \left.\frac{\partial \mathcal{L}_p^\phi(\mathbf{F})}{\partial \mathbf{F}_{i,:}}\right|_{\mathbf{F}^{(*)}} \tag{26}$$

For all $i, j \in [N]$, let $\mathbf{v} \in \mathbb{R}^{1 \times c}$ be a disturbance acting on node $i$. Define the following:

$$N_{i,j}^{(k)} = W_{i,j} \left\| \sqrt{\frac{W_{i,j}}{D_{i,i}}} \mathbf{F}_{i,:}^{(k)} - \sqrt{\frac{W_{i,j}}{D_{j,j}}} \mathbf{F}_{j,:}^{(k)} \right\|^{p-2}$$

$$N_{i,j}'^{(k)} = W_{i,j} \left\| \sqrt{\frac{W_{i,j}}{D_{i,i}}} (\mathbf{F}_{i,:}^{(k)} + \mathbf{v}) - \sqrt{\frac{W_{i,j}}{D_{j,j}}} \mathbf{F}_{j,:}^{(k)} \right\|^{p-2}$$

$$M_{i,j}^{(k)} = N_{ij}^{(k)} \zeta_{i,j}(\mathbf{F}^{(k)}), \quad M_{i,j}'^{(k)} = N_{ij}'^{(k)} \zeta_{i,j}(\mathbf{F}^{(k)} + \mathbf{v}) \tag{27}$$

$$\alpha_{ii}'^{(k)} = 1 / \left( \sum_{v_j \sim v_i} \frac{M_{i,j}'^{(k)}}{D_{i,i}} + 2\mu \right), \qquad \beta_{ii}'^{(k)} = 2\mu \alpha_{ii}'^{(k)}$$

$$\mathbf{F}_{i,:}'^{(k+1)} = \alpha_{i,i}'^{(k)} \sum_{v_j \sim v_i} \frac{\mathbf{M}_{i,j}'^{(k)}}{\sqrt{D_{i,i} D_{j,j}}} \mathbf{F}_{j,:}^{(k)} + \beta'^{(k)} \mathbf{Y}_{i,:},$$

where $\zeta_{ij}(\mathbf{F})$ is defined as eq. (20) and $\mathbf{F}^{(k)} + \mathbf{v}$ means that $\mathbf{v}$ only applies to the $i$-th of $\mathbf{F}^{(k)}$ [2].
Similar to eq. (25), we compute

$$\partial \mathcal{L}_p^\phi(\mathbf{F}_{i,:}^{(k)} + \mathbf{v}) = \frac{1}{\alpha_{i,i}'^{(k)}} \left( (\mathbf{F}_{i,:}^{(k)} + \mathbf{v}) - \mathbf{F}_{i,:}'^{(k+1)} \right). \tag{28}$$

Hence from both eq. (25) and eq. (28) we will have

$$\left\| \partial \mathcal{L}_p^\phi(\mathbf{F}_{i,:}^{(k)} + \mathbf{v}) - \partial \mathcal{L}_p^\phi(\mathbf{F}_{i,:}^{(k)}) \right\| = \left\| \frac{1}{\alpha_{i,i}'^{(k)}} \left( (\mathbf{F}_{i,:}^{(k)} + \mathbf{v}) - \mathbf{F}_{i,:}'^{(k+1)} \right) - \frac{1}{\alpha_{i,i}^{(k)}} \left( \mathbf{F}_{i,:}^{(k)} - \mathbf{F}_{i,:}^{(k+1)} \right) \right\|$$

$$\leq \frac{1}{\alpha_{i,i}'^{(k)}} \|\mathbf{v}\| + \left\| \frac{1}{\alpha_{i,i}'^{(k)}} \left( \mathbf{F}_{i,:}^{(k)} - \mathbf{F}_{i,:}'^{(k+1)} \right) - \frac{1}{\alpha_{i,i}^{(k)}} \left( \mathbf{F}_{i,:}^{(k)} - \mathbf{F}_{i,:}^{(k+1)} \right) \right\|$$

$$= \frac{1}{\alpha_{i,i}'^{(k)}} \|\mathbf{v}\| + \left\| \left( \frac{1}{\alpha_{i,i}'^{(k)}} - \frac{1}{\alpha_{i,i}^{(k)}} \right) \mathbf{F}_{i,:}^{(k)} - \frac{1}{\alpha_{i,i}'^{(k)}} \mathbf{F}_{i,:}'^{(k+1)} + \frac{1}{\alpha_{i,i}^{(k)}} \mathbf{F}_{i,:}^{(k+1)} \right\|$$

$$= \frac{1}{\alpha_{i,i}'^{(k)}} \|\mathbf{v}\| + \left\| \sum_{v_j \sim v_i} \left( \frac{M_{i,j}'^{(k)}}{D_{i,i}} - \frac{M_{i,j}^{(k)}}{D_{i,i}} \right) \mathbf{F}_{i,:}^{(k)} - \sum_{v_j \sim v_i} \left( \frac{M_{i,j}'^{(k)}}{\sqrt{D_{i,i} D_{j,j}}} \right) \mathbf{F}_{j,:}'^{(k)} + \left( \frac{M_{i,j}^{(k)}}{\sqrt{D_{i,i} D_{j,j}}} \right) \mathbf{F}_{j,:}^{(k)} \right\|$$

$$= \left( \sum_{v_j \sim v_i} \frac{M_{i,j}^{(k)}}{D_{i,i}} + 2\mu \right) \|\mathbf{v}\| + \sum_{v_j \sim v_i} \left( \frac{M_{i,j}'^{(k)} - M_{i,j}^{(k)}}{D_{i,i}} \right) \|\mathbf{v}\|$$

$$+ \left\| \sum_{v_j \sim v_i} \left( \frac{M_{i,j}'^{(k)} - M_{i,j}^{(k)}}{D_{i,i}} \right) \mathbf{F}_{i,:}^{(k)} - \sum_{v_j \sim v_i} \left( \frac{M_{i,j}'^{(k)} - M_{i,j}^{(k)}}{\sqrt{D_{i,i} D_{j,j}}} \right) \mathbf{F}_{j,:}^{(k)} \right\|.$$

Note that in eq. (27), $\| \cdot \|^{p-2}$ is the matrix $L_2$ norm raised to power $p-2$, that is $\|\mathbf{X}\|^{p-2} = \left( \left( \sum_{i,j} x_{i,j}^2 \right)^{\frac{1}{2}} \right)^{p-2}$.
It is known that the matrix $L_2$ norm as a function is Lipschitz (Paulavičius & Žilinskas, 2006), so is its exponential to $p-2$. Furthermore, it is easy to verify that $\|\mathbf{N}' - \mathbf{N}\| \leq c\|\mathbf{v}\|$ due to the property of $\mathbf{N}$ and $\mathbf{N}'$. Hence, according to Lemma 1, the following holds

$$|M_{i,j}'^{(k)} - M_{i,j}^{(k)}| \leq C|N_{i,j}'^{(k)} - N_{i,j}^{(k)}| \leq C'\|\mathbf{v}\|.$$

---

[2] With slightly abuse of notation, we denote $N'$ as the matrix after assigning the disturbance $\mathbf{v}$ to the matrix $N$.

Combining all the above, we have

$$\left\| \partial \mathcal{L}_p^\phi(\mathbf{F}_{i,:}^{(k)} + \mathbf{v}) - \partial \mathcal{L}_p^\phi(\mathbf{F}_{i,:}^{(k)}) \right\| \le \left( \sum_{v_j \sim v_i} \frac{M_{i,j}^{(k)}}{D_{i,i}} + 2\mu + o(\mathcal{G}, \mathbf{v}, \mathbf{X}, p) \right) \|\mathbf{v}\|, \tag{29}$$

where $o(\mathcal{G}, \mathbf{v}, \mathbf{X}, p)$ is bounded. It is worth noting that the quantity of $o(\mathcal{G}, \mathbf{v}, \mathbf{X}, p)$ is bounded by

$$\sum_{v_j \sim v_i} \left( \frac{M_{i,j}'^{(k)} - M_{i,j}^{(k)}}{D_{i,i}} \right) \|\mathbf{v}\| + \left\| \sum_{v_j \sim v_i} \left( \frac{M_{i,j}'^{(k)} - M_{i,j}^{(k)}}{D_{i,i}} \right) \mathbf{F}_{i,:}^{(k)} - \sum_{v_j \sim v_i} \left( \frac{M_{i,j}'^{(k)} - M_{i,j}^{(k)}}{\sqrt{D_{i,i} D_{j,j}}} \right) \mathbf{F}_{j,:}^{(k)} \right\|.$$

Let $\overline{o} = o(\mathcal{G}, \mathbf{v}, \mathbf{X}, p)$, $\boldsymbol{\gamma} = \{\gamma_1, ... \gamma_N\}^\top$, and $\boldsymbol{\eta} \in \mathbb{R}^{N \times c}$. By the Taylor expansion theorem we have:

$$\mathcal{L}_p^\phi(\mathbf{F}_{i,:}^{(k)} + \gamma_i \boldsymbol{\eta}_{i,:}) = \mathcal{L}_p^\phi(\mathbf{F}_{i,:}^{(k)}) + \gamma_i \int_0^1 \langle \partial \mathcal{L}_p^\phi(\mathbf{F}_{i,:}^{(k)} + \epsilon \gamma_i \boldsymbol{\eta}_{i,:}), \boldsymbol{\eta}_{i,:} \rangle d\epsilon \quad \forall i$$

$$= \mathcal{L}_p^\phi(\mathbf{F}_{i,:}^{(k)}) + \langle \partial \mathcal{L}_p^\phi(\mathbf{F}_{i,:}^{(k)}), \boldsymbol{\eta}_{i,:} \rangle + \gamma_i \int_0^1 \left\langle \partial \mathcal{L}_p^\phi \left( \mathbf{F}_{i,:}^{(k)} + \epsilon \gamma_i \boldsymbol{\eta}_{i,:} - \partial \mathcal{L}_p^\phi \left( \mathbf{F}_{i,:}^{(k)} \right) \right), \boldsymbol{\eta}_{i,:} \right\rangle d\epsilon$$

$$\le \mathcal{L}_p^\phi(\mathbf{F}_{i,:}^{(k)}) + \langle \partial \mathcal{L}_p^\phi(\mathbf{F}_{i,:}^{(k)}), \boldsymbol{\eta}_{i,:} \rangle \gamma_i + \gamma_i \int_0^1 \left\| \partial \mathcal{L}_p^\phi \left( \mathbf{F}_{i,:}^{(k)} + \epsilon \gamma_i \boldsymbol{\eta}_{i,:} - \partial \mathcal{L}_p^\phi \left( \mathbf{F}_{i,:}^{(k)} \right) \right) \right\| \|\boldsymbol{\eta}_{i,:}\| d\epsilon$$

$$\le \mathcal{L}_p^\phi(\mathbf{F}_{i,:}^{(k)}) + \langle \partial \mathcal{L}_p^\phi(\mathbf{F}_{i,:}^{(k)}), \boldsymbol{\eta}_{i,:} \rangle \gamma_i + \left( \frac{1}{\alpha_{i,i}^{(k)}} + \overline{o} \right) \gamma_i^2 \|\boldsymbol{\eta}_{i,:}\|^2$$

where the last inequality comes from eq. (29).

Taking $\gamma_i = \alpha_{ii}^{(k)}$ and $\boldsymbol{\eta}_{i,:} = -\partial \mathcal{L}_p^\phi(\mathbf{F}_{i,:}^{(k)})$ in the above inequality gives

$$\mathcal{L}_p^\phi \left( \mathbf{F}_{i,:}^{(k)} - \alpha_{ii}^{(k)} \partial \mathcal{L}_p^\phi(\mathbf{F}_{i,:}^{(k)}) \right)$$

$$\le \mathcal{L}_p^\phi(\mathbf{F}_{i,:}^{(k)}) - \alpha_{ii}^{(k)} \left\langle \partial \mathcal{L}_p^\phi(\mathbf{F}_{i,:}^{(k)}), \partial \mathcal{L}_p^\phi(\mathbf{F}_{i,:}^{(k)}) \right\rangle + \frac{1}{2} \left( \frac{1}{\alpha_{i,i}^{(k)}} + \overline{o} \right) \alpha_{i,i}^{2(k)} \|\partial \mathcal{L}_p^\phi(\mathbf{F}_{i,:}^{(k)})\|^2$$

$$= \mathcal{L}_p^\phi(\mathbf{F}_{i,:}^{(k)}) - \frac{1}{2} \alpha_{i,i}^{(k)} \left( 1 - \alpha_{i,i}^{(k)} \overline{o} \right) \|\partial \mathcal{L}_p^\phi(\mathbf{F}_{i,:}^{(k)})\|^2. \tag{30}$$

Given that $\overline{o}$ is bounded, if we choose a large $\mu$, e.g., $2\mu > \overline{o}$, we will have

$$1 - \alpha_{i,i}^{(k)} \overline{o} = 1 - \frac{\overline{o}}{\sum_{v_j \sim v_i} \frac{M_{i,j}^{(k)}}{D_{i,i}} + 2\mu} > 0.$$

Thus the second term in eq. (30) is positive. Hence we have

$$\mathcal{L}_p^\phi(\mathbf{F}_{i,:}^{(k+1)}) := \mathcal{L}_p^\phi \left( \mathbf{F}_{i,:}^{(k)} - \alpha_{ii}^{(k)} \partial \mathcal{L}_p^\phi(\mathbf{F}_{i,:}^{(k)}) \right) \le \mathcal{L}_p^\phi(\mathbf{F}_{i,:}^{(k)}).$$

This completes the proof. $\qquad \square$

Theorem 1 shows that with an appropriately chosen value of $\mu$, the iteration scheme for the implicit layer eq. (18) is guaranteed to coverage. This inspires us to explore further on the variation of the node feature produced from implicit layer asymptotically. Recall that to measure the difference between node features, one common choice is to analyze its Dirichlet energy, which is initially considered in the setting $p = 2$ in eq. (15). It is known that the Dirichlet energy of the node feature tend to approach to 0 after sufficiently large number of iterations in many GNN models (Kipf & Welling, 2016; Wu et al., 2020; Chamberlain et al., 2021a; Di Giovanni et al., 2022), known as over-smoothing problem. However, as we will show in the next section, by taking large $\mu$ or small $p$, the iteration from the implicit layer will always lift up the Dirichlet energy of the node features, and over-smoothing issue can be resolved completely in pL-UFG.

### 3.2 Energy Behavior of the pL-UFG

In this section, we show the energy behavior of the p-Laplacian based implicit layer. Specifically, we are interested in analyzing the property of the generalized Dirichlet energy defined in (Bronstein et al., 2021).We start by denoting generalized graph convolution as follows:

$$\mathbf{F}^{(k+\tau)} = \mathbf{F}^{(k)} + \tau\sigma\left(-\mathbf{F}^{(k)}\mathbf{\Omega}^{(k)} + \widehat{\mathbf{A}}\mathbf{F}^{(k)}\widehat{\mathbf{W}}^{(k)} - \mathbf{F}^{(0)}\widetilde{\mathbf{W}}^{(k)}\right), \tag{31}$$

where $\mathbf{\Omega}^{(k)}, \widehat{\mathbf{W}}^{(k)}$ and $\widetilde{\mathbf{W}}^{(k)} \in \mathbb{R}^{c\times c}$ act on each node feature vector independently and perform channel mixing. When $\tau = 1$, and $\mathbf{\Omega}^{(k)} = \widetilde{\mathbf{W}}^{(k)} = \mathbf{0}$, it returns to GCN (Kipf & Welling, 2016). Additionally, by setting $\mathbf{\Omega}^{(k)} \neq 0$, we have the anisotropic instance of GraphSAGE (Xu et al., 2019). To quantify the quality of the node features generated by eq. (31), specifically, Bronstein et al. (2021) considered a new class of energy as defined below,

$$\mathbf{E}(\mathbf{F}) = \frac{1}{2}\sum_{i=1}^{N}\langle\mathbf{f}_i, \mathbf{\Omega}\mathbf{f}_i\rangle - \frac{1}{2}\sum_{i,j=1}^{N}\widehat{\mathbf{A}}_{i,j}\langle\mathbf{f}_i, \widehat{\mathbf{W}}\mathbf{f}_j\rangle + \varphi^{(0)}(\mathbf{F}, \mathbf{F}^{(0)}), \tag{32}$$

in which $\varphi^{(0)}(\mathbf{F}, \mathbf{F}^{(0)})$ serves a function of that induces the source term from $\mathbf{F}$ or $\mathbf{F}^{(0)}$. It is worth noting that by setting $\mathbf{\Omega} = \widehat{\mathbf{W}} = \mathbf{I}_c$ and $\varphi^{(0)} = 0$, we recover the classic Dirichlet energy when setting $p = 2$ in eq. (15) that is, $\mathbf{E}(\mathbf{F}) = \frac{1}{2}\sum_{(v_i, v_j)\in\mathcal{E}}\left\|\sqrt{\frac{w_{i,j}}{d_{j,j}}}\mathbf{f}_j - \sqrt{\frac{w_{i,j}}{d_{i,i}}}\mathbf{f}_i\right\|^2$. Additionally, when we set $\varphi^{(0)}(\mathbf{F}, \mathbf{F}^{(0)}) = \sum_i\langle\mathbf{f}_i, \widetilde{\mathbf{W}}\mathbf{f}_i^{(0)}\rangle$, eq. (32) can be rewritten as:

$$\mathbf{E}(\mathbf{F}) = \left\langle\mathrm{vec}(\mathbf{F}), \frac{1}{2}(\mathbf{\Omega}\otimes\mathbf{I}_N - \widehat{\mathbf{W}}\otimes\widehat{\mathbf{A}})\mathrm{vec}(\mathbf{F}) + (\widetilde{\mathbf{W}}\otimes\mathbf{I}_N)\mathrm{vec}(\mathbf{F}^{(0)})\right\rangle. \tag{33}$$

Recall that eq. (19) produces the node feature $\mathbf{F}^{(k+1)}$ according to the edge diffusion $\boldsymbol{\alpha}\mathbf{D}^{-1/2}\mathbf{M}\mathbf{D}^{-1/2}$ on $\mathbf{F}^{(k)}$ and the scaled source term $2\mu\boldsymbol{\alpha}\mathbf{F}^{(0)}$ where $\mathbf{F}^{(0)}$ can be set to $\mathbf{Y}$. To be specific, in (33), we set $\mathbf{\Omega} = \widehat{\mathbf{W}} = \widetilde{\mathbf{W}} = \mathbf{I}_c$ and replace the edge *diffusion* $\widehat{\mathbf{A}}$ with $\boldsymbol{\alpha}\mathbf{D}^{-1/2}\mathbf{M}\mathbf{D}^{-1/2}$ and set the identity matrix $\mathbf{I}_N$ in the residual term to be the diagonal matrix $2\mu\boldsymbol{\alpha}$. Finally we propose

**Definition 7** (The Generalized Dirichlet Energy). *Based on the previous notation setting, the generalized Dirichlet energy for the node features* $\mathbf{F}^{(k+1)}$ *in eq. (19) is:*

$$\mathbf{E}^{PF}(\mathbf{F}^{(k+1)}) = \left\langle\mathrm{vec}(\mathbf{F}^{(k+1)}),\right.$$
$$\left.\frac{1}{2}\left(\mathbf{I}_c\otimes\mathbf{I}_N - \mathbf{I}_c\otimes\left(\boldsymbol{\alpha}^{(k+1)}\mathbf{D}^{-1/2}\mathbf{M}^{(k+1)}\mathbf{D}^{-1/2}\right)\right)\mathrm{vec}(\mathbf{F}^{(k+1)}) + (\mathbf{I}_c\otimes 2\mu\boldsymbol{\alpha}^{(k+1)})\mathrm{vec}(\mathbf{F}^{(0)})\right\rangle, \tag{34}$$

*where the superscript "PF" is short for p-Laplacian based framelet models.*

It is worth noting that the generalized Dirichlet energy defined in eq. (34) is dynamic along the iterative layers due to the non-linear nature of the implicit layer defined in eq. (18). We are now able to analyze the energy ($\mathbf{E}^{PF}(\mathbf{F})$) behavior of the pL-UFG, concluded as the following proposition.

**Proposition 2** (Energy Behavior). *Assume* $\mathcal{G}$ *is connected, unweighted and undirected. There exists sufficiently large value of* $\mu$ *or small value of* $p$ *such that* $\mathbf{E}^{PF}(\mathbf{F})$ *will stay away above 0 at each iterative layer* $k$ *and increases with the increase of* $\mu$ *or the decrease of* $p$.

*Proof.* We start with the definition of the generalized Dirichlet energy above, we can re-write $\mathbf{E}^{PF}(\mathbf{F}^{(k+1)})$ in the following inner product between $\mathrm{vec}(\mathbf{F}^{(k+1)})$ and $\mathrm{vec}(\mathbf{F}^{(0)})$, based on $\mathbf{M}$, $\boldsymbol{\alpha}$, $\boldsymbol{\beta}$ and the iterative scheme

defined in eq. (19):

$$
\begin{aligned}
\mathbf{E}^{PF}(\mathbf{F}^{(k+1)}) = &\Big\langle \mathrm{vec}(\mathbf{F}^{(k+1)}), \\
&\frac{1}{2}\Big(\mathbf{I}_c \otimes \mathbf{I}_N - \mathbf{I}_c \otimes \big(\boldsymbol{\alpha}^{(k+1)}\mathbf{D}^{-1/2}\mathbf{M}^{(k+1)}\mathbf{D}^{-1/2}\big)\Big)\mathrm{vec}(\mathbf{F}^{(k+1)}) + (\mathbf{I}_c \otimes 2\mu\boldsymbol{\alpha}^{(k+1)})\mathrm{vec}(\mathbf{F}^{(0)})\Big\rangle \\
= &\Big\langle \mathrm{vec}(\mathbf{F}^{(k+1)}), \mathrm{vec}(\mathbf{F}^{(k+1)})\Big\rangle - \frac{1}{2}\Big\langle \mathrm{vec}(\mathbf{F}^{(k+1)}), \mathbf{I}_c \otimes \big(\boldsymbol{\alpha}^{(k+1)}\mathbf{D}^{-1/2}\mathbf{M}^{(k+1)}\mathbf{D}^{-1/2}\big)\mathrm{vec}(\mathbf{F}^{(k+1)}) \\
&-(\mathbf{I}_c \otimes 4\mu\boldsymbol{\alpha}^{(k+1)})\mathrm{vec}(\mathbf{F}^{(0)})\Big\rangle.
\end{aligned}
\tag{35}
$$

Based on the form of eq. (35), it is straightforward to see that to let $\mathbf{E}^{PF}(\mathbf{F}^{(k+1)}) > 0$ and further increase with the desired quantities of $\mu$ and $p$, it is sufficient to require[3]:

$$
\mathbf{I}_c \otimes \big(\boldsymbol{\alpha}^{(k+1)}\mathbf{D}^{-1/2}\mathbf{M}^{(k+1)}\mathbf{D}^{-1/2}\big)\mathrm{vec}(\mathbf{F}^{(k+1)}) - (\mathbf{I}_c \otimes 2\mu\boldsymbol{\alpha}^{(k+1)})\mathrm{vec}(\mathbf{F}^{(0)}) < 0.
\tag{36}
$$

To explicitly show how the quantities of $\mu$ and $p$ affect the term in eq. (36), we start with the case when $k = 0$. When $k = 0$, eq. (36) becomes:

$$
\begin{aligned}
&\mathbf{I}_c \otimes \big(\boldsymbol{\alpha}^{(1)}\mathbf{D}^{-1/2}\mathbf{M}^{(1)}\mathbf{D}^{-1/2}\big)\mathrm{vec}(\mathbf{F}^{(1)}) - (\mathbf{I}_c \otimes 2\mu\boldsymbol{\alpha}^{(1)})\mathrm{vec}(\mathbf{F}^{(0)}) \\
=&\mathbf{I}_c \otimes \big(\boldsymbol{\alpha}^{(1)}\mathbf{D}^{-1/2}\mathbf{M}^{(1)}\mathbf{D}^{-1/2}\big)\mathrm{vec}\big(\boldsymbol{\alpha}^{(0)}\mathbf{D}^{-1/2}\mathbf{M}^{(0)}\mathbf{D}^{-1/2}\mathbf{F}^{(0)} + 2\mu\boldsymbol{\alpha}^{(0)}\mathbf{F}^{(0)}\big) - (\mathbf{I}_c \otimes 2\mu\boldsymbol{\alpha}^{(1)})\mathrm{vec}(\mathbf{F}^{(0)}), \\
=&\mathbf{I}_c \otimes \big(\boldsymbol{\alpha}^{(1)}\mathbf{D}^{-1/2}\mathbf{M}^{(1)}\mathbf{D}^{-1/2}\big)\Big(\mathbf{I}_c \otimes \big(\boldsymbol{\alpha}^{(0)}\mathbf{D}^{-1/2}\mathbf{M}^{(0)}\mathbf{D}^{-1/2} + 2\mu\boldsymbol{\alpha}^{(0)}\big)\mathrm{vec}(\mathbf{F}^{(0)})\Big) - (\mathbf{I}_c \otimes 2\mu\boldsymbol{\alpha}^{(1)})\mathrm{vec}(\mathbf{F}^{(0)}), \\
=&\mathbf{I}_c \otimes \Big(\prod_{s=0}^{1} \boldsymbol{\alpha}^{(s)}\mathbf{D}^{-1/2}\mathbf{M}^{(s)}\mathbf{D}^{-1/2} + \big(\boldsymbol{\alpha}^{(1)}\mathbf{D}^{-1/2}\mathbf{M}^{(1)}\mathbf{D}^{-1/2}2\mu\boldsymbol{\alpha}^{(0)}\big) - 2\mu\boldsymbol{\alpha}^{(1)}\Big)\mathrm{vec}(\mathbf{F}^{(0)}).
\end{aligned}
\tag{37}
$$

We note that, in eq. (37), $\Big(\prod_{s=0}^{1} \boldsymbol{\alpha}^{(s)}\mathbf{D}^{-1/2}\mathbf{M}^{(s)}\mathbf{D}^{-1/2} + \big(\boldsymbol{\alpha}^{(1)}\mathbf{D}^{-1/2}\mathbf{M}^{(1)}\mathbf{D}^{-1/2}2\mu\boldsymbol{\alpha}^{(0)}\big) - 2\mu\boldsymbol{\alpha}^{(1)}\Big)$ can be computed as:

$$
\begin{aligned}
&\prod_{s=0}^{1} \alpha_{i,i}^{(s)}d_{i,i}^{-1/2}M_{i,j}^{(s)}d_{j,j}^{-1/2} + \Big(\alpha_{i,i}^{(1)}d_{i,i}^{-1/2}M_{i,j}^{(1)}d_{j,j}^{-1/2}2\mu\alpha_{i,i}^{(0)}\Big) - 2\mu\alpha_{i,i}^{(1)} \\
=&\prod_{s=0}^{1}\left[\left(1/\left(\sum_{v_j \sim v_i}\frac{M_{i,j}^{(s)}}{d_{i,i}} + 2\mu\right)\right)\left(\frac{\big\|\nabla_W\mathbf{F}^{(s)}([i,j])\big\|^{p-2}}{\sqrt{d_{i,i}d_{j,j}}}\right)\right] + \\
&\left[\left(1/\left(\sum_{v_j \sim v_i}\frac{M_{i,j}^{(1)}}{d_{i,i}} + 2\mu\right)\right)\left(\frac{\big\|\nabla_W\mathbf{F}^{(1)}([i,j])\big\|^{p-2}}{\sqrt{d_{i,i}d_{j,j}}}\right)\left(2\mu/\left(\sum_{v_j \sim v_i}\frac{M_{i,j}^{(0)}}{d_{i,i}} + 2\mu\right)\right)\right] \\
&-\left(2\mu/\left(\sum_{v_j \sim v_i}\frac{M_{i,j}^{(1)}}{d_{i,i}} + 2\mu\right)\right), \\
=&\left(\frac{\big\|\nabla_W\mathbf{F}^{(0)}([i,j])\big\|^{p-2}}{\Big(\sum_{v_j \sim v_i}\frac{M_{i,j}^{(0)}}{d_{i,i}} + 2\mu\Big)\cdot\sqrt{d_{i,i}d_{j,j}}}\right)\left(\frac{\big\|\nabla_W\mathbf{F}^{(1)}([i,j])\big\|^{p-2}}{\Big(\sum_{v_j \sim v_i}\frac{M_{i,j}^{(1)}}{d_{i,i}} + 2\mu\Big)\cdot\sqrt{d_{i,i}d_{j,j}}}\right) + \\
&\left(\frac{\big\|\nabla_W\mathbf{F}^{(1)}([i,j])\big\|^{p-2}}{\Big(\sum_{v_j \sim v_i}\frac{M_{i,j}^{(1)}}{d_{i,i}} + 2\mu\Big)\cdot\sqrt{d_{i,i}d_{j,j}}}\cdot 2\mu/\left(\sum_{v_j \sim v_i}\frac{M_{i,j}^{(0)}}{d_{i,i}} + 2\mu\right)\right) - \left(2\mu/\left(\sum_{v_j \sim v_i}\frac{M_{i,j}^{(1)}}{d_{i,i}} + 2\mu\right)\right).
\end{aligned}
\tag{38}
$$

---

[3]Strictly speaking, one shall further require all elements in $\mathbf{F}^{(k+1)}$ larger than or equal to 0. As this can be achieved by assigned a non-linear activation function (i.e., ReLU) to the framelet, we omit it here in our main analysis.

Now we see that by assigning a sufficient large of $\mu$ or small value of $p$, we can see terms like $\dfrac{\left\|\nabla_W \mathbf{F}^{(1)}([i,j])\right\|^{p-2}}{\left(\sum_{v_j \sim v_i} \frac{M_{i,j}^{(1)}}{d_{i,i}} + 2\mu\right) \cdot \sqrt{d_{i,i}d_{j,j}}}$ in eq. (38) are getting smaller Additionally, we have both $2\mu/\left(\sum_{v_j \sim v_i} \frac{M_{i,j}^{(0)}}{d_{i,i}} + 2\mu\right)$ and $2\mu/\left(\sum_{v_j \sim v_i} \frac{M_{i,j}^{(1)}}{d_{i,i}} + 2\mu\right) \approx 1$. Therefore, the summation result of eq. (38) tends to be negative. Based on eq. (35), $\mathbf{E}^{PF}(\mathbf{F}^{(k+1)})$ will stay above 0.

For the case that $k \geq 1$, by taking into the iterative algorithm eq. (19), eq. (37) becomes:

$$\mathbf{I}_c \otimes \left(\left(\prod_{s=0}^{k+1} \boldsymbol{\alpha}^{(s)} \mathbf{D}^{-1/2}\mathbf{M}^{(s)}\mathbf{D}^{-1/2} + \sum_{s=0}^{k+1}\left(\prod_{l=k-s}^{k+1} \boldsymbol{\alpha}^{(l)}\mathbf{D}^{-1/2}\mathbf{M}^{(l)}\mathbf{D}^{-1/2}\right)\left(2\mu\boldsymbol{\alpha}^{(l-1)}\right) - 2\mu\boldsymbol{\alpha}^{(k+1)}\right)\right)\text{vec}(\mathbf{F}^{(0)}).$$

Applying the same reasoning as before, it is not hard to verify that with sufficient large of $\mu$ and small of $p$, the term $\left(\prod_{s=0}^{k+1} \boldsymbol{\alpha}^{(s)}\mathbf{D}^{-1/2}\mathbf{M}^{(s)}\mathbf{D}^{-1/2} + \sum_{s=0}^{k+1}\left(\prod_{l=k-s}^{k+1} \boldsymbol{\alpha}^{(l)}\mathbf{D}^{-1/2}\mathbf{M}^{(l)}\mathbf{D}^{-1/2}\right)\left(2\mu\boldsymbol{\alpha}^{(l-1)}\right) - 2\mu\boldsymbol{\alpha}^{(k+1)}\right)$ in the above equation tend to be negative, yielding a positive $\mathbf{E}^{PF}(\mathbf{F}^{(k+1)})$. Asymptotically, we have:

$$\mathbf{E}^{PF}(\mathbf{F}^{(k+1)}) \approx \left\langle \text{vec}(\mathbf{F}^{(k+1)}), \text{vec}(\mathbf{F}^{(k+1)})\right\rangle + \left\langle \text{vec}(\mathbf{F}^{(k+1)}), \frac{1}{2}\left(\mathbf{I}_c \otimes \left(4\mu\boldsymbol{\alpha}^{(k+1)} + \mathbf{I}_N\right)\right)\text{vec}(\mathbf{F}^{(0)})\right\rangle. \quad (39)$$

This shows that the energy increases along with the magnitude of $\mu$, and it is not hard to express eq. (39) as the similar form of eq. (38) and verify that the energy decreases with the quantity of $p$. This completes the proof. □

**Remark 3.** Proposition 2 shows that, for any of our framelet convolution models, the p-Laplacian based implicit layer will not generate identical node feature across graph nodes, and thus the so-called over-smoothing issue will not appear **asymptotically**. Furthermore, it is worth noting that the result from Proposition 2 provides the theoretical justification of the empirical observations in (Shao et al., 2022), where a large value of $\mu$ or small value of $p$ is suitable for fitting heterophily datasets which commonly require the output of GNN to have higher Dirichlet energy.

**Remark 4** (Regarding to the quantity of $p$)**.** The conclusion of Proposition 2 is under sufficient large of $\mu$ or small of $p$. However, it is well-known that the quantity of $p$ cannot be set as arbitrary and in fact it is necessary to have $p \geq 1$ so that the iteration for the solution of the optimization problem defined in eq. (18) can converge. Therefore, it is not hard to see that the effect of $p$ is weaker than $\mu$ in terms of analyzing the asymptotic behavior of the model (i.e., via eq. (38)). Without loss of generality, in the sequel, when we analyze the property of the model with conditions on $\mu$ and $p$, we mainly target on the effect from $\mu$ and one can check from eq. (38) $\mu$ and $p$ are with opposite effect on the model.

## 3.3 Interaction with Framelet Energy Dynamic

To analyze the interaction between the energy dynamic of framelet convolution defined in eq. (9) and the p-Laplacian based implicit propagation (Shao et al., 2022), We first briefly review some recent work on the energy dynamic of the GNNs. In (Di Giovanni et al., 2022), the propagation of GNNs was considered as the gradient flow of the Dirichlet energy that can be formulated as:

$$\mathbf{E}(\mathbf{F}) = \frac{1}{2}\sum_{i=1}^{N}\sum_{j=1}^{N}\left\|\sqrt{\frac{w_{i,j}}{d_{j,j}}}\mathbf{f}_j - \sqrt{\frac{w_{i,j}}{d_{i,i}}}\mathbf{f}_i\right\|^2, \quad (40)$$

and similarly by setting the power from 2 to $p$, we recover the p-Dirichlet form presented in eq. (15). The gradient flow of the Dirichlet energy yields the so-called graph heat equation (Chung, 1997) as $\dot{\mathbf{F}}^{(k)} = -\nabla\mathbf{E}(\mathbf{F}^{(k)}) = -\widetilde{\mathbf{L}}\mathbf{F}^{(k)}$. Its Euler discretization leads to the propagation of linear GCN models (Wu et al., 2019a; Wang et al., 2021). The process is called Laplacian smoothing (Li et al., 2018) and it converges to the kernel of $\widetilde{\mathbf{L}}$, i.e., $\ker(\widetilde{\mathbf{L}})$ as $k \to \infty$, resulting in non-separation of nodes with same degrees, known as the over-smoothing issue.

Following this observation, the work (Han et al., 2022; Di Giovanni et al., 2022) also show even with the help of the non-linear activation function and the weight matrix via classic GCN (eq. (1)), the process described is still dominated by the low frequency (small Laplacian eigenvalues) of the graph, hence eventually converging to the kernel of $\widetilde{\mathbf{L}}$, for almost every initialization. To quantify such behavior, Di Giovanni et al. (2022); Han et al. (2022) consider a general dynamic as $\dot{\mathbf{F}}^{(k)} = \mathrm{GNN}_\theta(\mathbf{F}^{(k)}, k)$, with $\mathrm{GNN}_\theta(\cdot)$ as an arbitrary graph neural network function, and also characterizes its behavior by low/high-frequency-dominance (L/HFD).

**Definition 8** ((Di Giovanni et al., 2022)). $\dot{\mathbf{F}}^{(k)} = \mathrm{GNN}_\theta(\mathbf{F}^{(k)}, k)$ *is Low-Frequency-Dominant (LFD) if* $\mathbf{E}\big(\mathbf{F}^{(k)}/\|\mathbf{F}^{(k)}\|\big) \to 0$ *as* $k \to \infty$, *and is High-Frequency-Dominant (HFD) if* $\mathbf{E}\big(\mathbf{F}^{(k)}/\|\mathbf{F}^{(k)}\|\big) \to \rho_{\widetilde{\mathbf{L}}}/2$ *as* $t \to \infty$.

**Lemma 2** ((Di Giovanni et al., 2022)). *A GNN model is LFD (resp. HFD) if and only if for each* $t_j \to \infty$, *there exists a sub-sequence indexed by* $k_{j_l} \to \infty$ *and* $\mathbf{F}_\infty$ *such that* $\mathbf{F}(k_{j_l})/\|\mathbf{F}(k_{j_l})\| \to \mathbf{F}_\infty$ *and* $\widetilde{\mathbf{L}}\mathbf{F}_\infty = 0$ *(resp.* $\widetilde{\mathbf{L}}\mathbf{F}_\infty = \rho_{\widetilde{\mathbf{L}}}\mathbf{F}_\infty$).

**Remark 5** (LFD, HFD and graph homophily). Based on Definition 8 and Lemma 2, for a given GNN model, if $\mathcal{G}$ is homophilic, i.e., adjacency nodes are more likely to share the same label, one may prefer for the model to induce a LFD dynamic in order to fit the characteristic of $\mathcal{G}$. On the other hand, if $\mathcal{G}$ is heterophilic, the model is expected to induce a HFD dynamic, so that even in the adjacent nodes, their predicted labels still tend to be different. Thus, ideally, a model should be flexible enough to accommodate both LFD and HFD dynamics.

Generalized from the energy dynamic framework provided in (Di Giovanni et al., 2022), Han et al. (2022) developed a framelet Dirichlet energy and analyzed the energy behavior of both spectral (eq. (9)) and spatial framelet (eq. (10)) convolutions. Specifically, let

$$\mathbf{E}^{Fr}(\mathbf{F}) = \frac{1}{2}\mathrm{Tr}\big((\mathcal{W}_{r,\ell}\mathbf{F})^\top \mathcal{W}_{r,\ell}\mathbf{F}\mathbf{\Omega}_{r,\ell}\big) - \frac{1}{2}\mathrm{Tr}\big((\mathcal{W}_{r,\ell}\mathbf{F})^\top \mathrm{diag}(\theta)_{r,\ell}\mathcal{W}_{r,\ell}\mathbf{F}\widehat{\mathbf{W}}\big)$$

for all $(r,\ell) \in \mathcal{I}$. The generated framelet energy is given by:

$$\begin{aligned}
\mathbf{E}^{Fr}(\mathbf{F}) &= \mathbf{E}^{Fr}_{0,J}(\mathbf{F}) + \sum_{r,\ell} \mathbf{E}^{Fr}_{r,\ell}(\mathbf{F}) \\
&= \frac{1}{2} \sum_{(r,\ell) \in \mathcal{I}} \Big\langle \mathrm{vec}(\mathbf{F}), \Big(\mathbf{\Omega}_{r,\ell} \otimes \mathcal{W}^\top_{r,\ell}\mathcal{W}_{r,\ell} - \widehat{\mathbf{W}} \otimes \mathcal{W}^\top_{r,\ell}\mathrm{diag}(\theta)_{r,\ell}\mathcal{W}_{r,\ell}\Big)\mathrm{vec}(\mathbf{F}) \Big\rangle,
\end{aligned} \tag{41}$$

where the superscript "$^{Fr}$" stands for the framelet convolution. This definition is based on the fact that the total Dirichlet energy is conserved under framelet decomposition (Han et al., 2022; Di Giovanni et al., 2022). By analyzing the gradient flow of the framelet energy [4] defined above, Han et al. (2022) concluded the energy dynamic of framelet as:

**Proposition 3** ((Han et al., 2022)). *The spectral graph framelet convolution eq. (9) with Haar-type filter (i.e.* $R = 1$ *in the case of scaling function set) can induce both LFD and HFD dynamics. Specifically, let* $\boldsymbol{\theta}_{0,\ell} = \mathbf{1}_N$ *and* $\boldsymbol{\theta}_{r,\ell} = \theta\mathbf{1}_N$ *for* $r = 1, ..., L, \ell = 1, ..., J$ *where* $\mathbf{1}_N$ *is a size* $N$ *vector of all* $1s$. *When* $\theta \in [0, 1)$, *the spectral framelet convolution is LFD and when* $\theta > 1$, *the spectral framelet convolution is HFD.*

It is worth noting that there are many other settings rather than $\boldsymbol{\theta}_{0,\ell} = \mathbf{1}_N$ and $\boldsymbol{\theta}_{r,\ell} = \theta\mathbf{1}_N$, i.e. adjusting $\theta$, for inducing LFD/HFD from framelet. However, in this paper, we only consider the conditions described in Proposition 3. To properly compare the energy dynamics between the framelet models, we present the following definition.

**Definition 9** (Stronger/Weaker Dynamic). *Let* $\mathcal{Q}_\theta$ *be the family of framelet models with the settings described in Proposition 3 and choice of* $\theta$. *We say that one framelet model* $\mathcal{Q}_{\theta_1}$ *is with a stronger LFD than another framelet model* $\mathcal{Q}_{\theta_2}$ *if* $\theta_1 < \theta_2$, *and weaker otherwise. Similarly, we say* $\mathcal{Q}_{\theta_1}$ *is with a stronger HFD than* $\mathcal{Q}_{\theta_2}$ *if* $\theta_1 > \theta_2$, *and weaker otherwise* [5].

---

[4]Similar to the requirement on our p-Laplacian based framelet energy ($\mathbf{E}^{PF}(\mathbf{F}^{(k+1)})$), to thoroughly verify the framelet energy in eq. (41) is a type of energy, we shall further require: $\nabla^2\mathbf{E}^{Fr}_{r,\ell}(\mathbf{F}) = \mathbf{\Omega}_{r,\ell} \otimes \mathcal{W}^\top_{r,\ell}\mathcal{W}_{r,\ell} - \widehat{\mathbf{W}} \otimes \mathcal{W}^\top_{r,\ell}\widetilde{\mathbf{L}}\mathcal{W}_{r,\ell}$ is symmetric, which can be satisfied by requiring both $\mathbf{\Omega}$ and $\widehat{\mathbf{W}}$ are symmetric.

[5]In case of any confusion, we note that in this paper we only compare the model's dynamics relationship when both of two (framelet) models are with the same frequency dominated dynamics (i.e., LFD, HFD).

**Remark 6.** Similar reasoning of Proposition 3 can be easily generalized to other commonly used framelet types such as Linear, Sigmoid and Entropy (Yang et al., 2022).

Before we present our conclusion, we note that to evaluate the changes of (framelet) energy behavior from the impact of implicit layer, one shall also define a layer-wised framelet energy such as $\mathbf{E}^{PF}(\mathbf{F}^{(k+1)})$ by only considering the energy from one step of propagation of graph framelet. With all these settings, we summarize the interaction between framelet and p-Laplacian based implicit propagation as:

**Lemma 3** (Stronger HFD). *Based on the condition described in Proposition 3, when framelet is HFD, with sufficient large value of $\mu$ or small of $p$, the p-Laplacian implicit propagation further amplify the energy $\mathbf{E}^{Fr}(\mathbf{F})$ in eq. (41) of the node feature (i.e., $\mathbf{Y}$ in eq. (18)) produced from the framelets, and thus achieving a higher HFD dynamic than original framelet in eq. (9).*

*Proof.* Recall that by setting sufficient large of $\mu$ or small of $p$, $\mathbf{E}^{PF}(\mathbf{F}^{(k+1)})$ in eq. (39) has the form

$$\mathbf{E}^{PF}(\mathbf{F}^{(k+1)}) \approx \left\langle \text{vec}(\mathbf{F}^{(k+1)}), \text{vec}(\mathbf{F}^{(k+1)}) \right\rangle + \left\langle \text{vec}(\mathbf{F}^{(k+1)}), \frac{1}{2}\left(\mathbf{I}_c \otimes \left(4\mu\boldsymbol{\alpha}^{(k+1)} + \mathbf{I}_N\right)\right)\text{vec}(\mathbf{F}^{(0)}) \right\rangle.$$

Similarly, when framelet is HFD, with $\boldsymbol{\theta}_{0,\ell} = \mathbf{1}_N$, $\boldsymbol{\theta}_{r,\ell} = \theta\mathbf{1}_N$ and $\theta > 1$, the Dirichlet energy (of $\mathbf{F}^{(k+1)}$) eq. (41) can be rewritten as:

$$\mathbf{E}^{Fr}(\mathbf{F}^{(k+1)}) = \frac{1}{2} \sum_{(r,\ell)\in\mathcal{I}} \left\langle \text{vec}(\mathbf{F}^{(k+1)}), \left(\boldsymbol{\Omega}_{r,\ell} \otimes \mathcal{W}_{r,\ell}^\top\mathcal{W}_{r,\ell} - \widehat{\mathbf{W}} \otimes \mathcal{W}_{r,\ell}^\top\text{diag}(\theta)_{r,\ell}\mathcal{W}_{r,\ell}\right)\text{vec}(\mathbf{F}^{(k+1)}) \right\rangle,$$

$$= \frac{1}{2} \sum_{(r,\ell)\in\mathcal{I}} \left\langle \text{vec}(\mathbf{F}^{(k+1)}), \left(\widehat{\mathbf{W}} \otimes \left(\mathcal{W}_{r,\ell}^\top\mathcal{W}_{r,\ell} - \mathcal{W}_{r,\ell}^\top\text{diag}(\theta)_{r,\ell}\mathcal{W}_{r,\ell}\right)\right)\text{vec}(\mathbf{F}^{(k+1)}) \right\rangle, \quad (42)$$

where the last equality is achieved by letting $\boldsymbol{\Omega} = \widehat{\mathbf{W}}$, meaning that no external force [6] exist within the space that contains the node features. We note that it is reasonable to have such assumption in order to explicitly analyze the energy changes in eq. (41) via the changes of $\theta$. Now we take the Haar framelet with $\ell = 1$ as an example, meaning there will be only one high-pass and low-pass frequency domain in the framelet model. Specifically, the R.H.S of eq. (42) can be further rewritten as:

$$\frac{1}{2} \sum_{(r,\ell)\in\mathcal{I}} \left\langle \text{vec}(\mathbf{F}^{(k+1)}), \left(\widehat{\mathbf{W}} \otimes \left(\mathcal{W}_{r,\ell}^\top\mathcal{W}_{r,\ell} - \mathcal{W}_{r,\ell}^\top\text{diag}(\theta)_{r,\ell}\mathcal{W}_{r,\ell}\right)\right)\text{vec}(\mathbf{F}^{(k+1)}) \right\rangle$$

$$\approx \frac{1}{2} \left\langle \text{vec}(\mathbf{F}^{(k+1)}), \left(\widehat{\mathbf{W}} \otimes \left(\mathbf{I}_N - \mathcal{W}_{1,1}^\top\text{diag}(\theta)_{1,1}\mathcal{W}_{1,1}\right)\right)\text{vec}(\mathbf{F}^{(k+1)}) \right\rangle. \quad (43)$$

The inclusion of $\mathcal{W}_{1,1}^\top\text{diag}(\theta)_{1,1}\mathcal{W}_{1,1}$ is based on the form of Haar type framelet with one scale. In addition, the approximation in eq. (43) is due to the outcome of HFD [7]. Now we combine the framelet energy in the above equation (eq. (43)) with the energy induced from p-Laplacian based implicit propagation (eq. (39)). Denote the total energy induced from framelet and implicit layer as:

$$\mathbf{E}^{(total)}(\mathbf{F}^{(k+1)}) = \left\langle \text{vec}(\mathbf{F}^{(k+1)}), \text{vec}(\mathbf{F}^{(k+1)}) \right\rangle \quad (44)$$

$$+ \frac{1}{2}\left\langle \text{vec}(\mathbf{F}^{(k+1)}), \left(\left(\widehat{\mathbf{W}} \otimes \left(\mathbf{I}_N - \mathcal{W}_{1,1}^\top\text{diag}(\theta)_{1,1}\mathcal{W}_{1,1}\right)\right)\text{vec}(\mathbf{F}^{(k+1)}) + \mathbf{I}_c \otimes \left(4\mu\boldsymbol{\alpha}^{(k+1)} + \mathbf{I}_N\right)\text{vec}(\mathbf{F}^{(0)})\right) \right\rangle.$$

It is not hard to check that $\mathbf{E}^{(total)}(\mathbf{F}^{(k+1)})$ is larger than $\mathbf{E}^{Fr}(\mathbf{F}^{(k+1)})$ (the framelet energy under HFD). Hence we have verified that the implicit layer further amplifies the Dirichlet energy. Moreover, one can approximate this stronger dynamic by re-parameterizing $\mathbf{E}^{(total)}(\mathbf{F}^{(k+1)})$ via assigning a higher quantity of $\theta' > \theta > 0$ and excluding the residual term. Hence, the inclusion of the implicit layer induces a higher HFD dynamic to framelet, and that completes the proof. □

---

[6]For details, please check (Bronstein et al., 2021)

[7]The result in eq. (43) provides identical conclusion on the claim in (Di Giovanni et al., 2022) such that in order to have a HFD dynamic, $\widehat{\mathbf{W}}$ must have negative eigenvalue(s).

**Corollary 1** (Escape from Over-smoothing). *With the same conditions in Proposition 3, when framelet is LFD, the implicit layer (with sufficient large $\mu$ or small $p$) ensures the Dirichlet energy of node features does not converge to 0, thus preventing the model from the over-smoothing issue.*

*Proof.* The proof can be done by combining Proposition 3 and Proposition 2, with the former illustrates that when model is HFD, there will be no over-smoothing problem, and the latter shows that even when the model is LFD, the Dirichlet energy of the node features will not converge to 0. Accordingly, pL-UFG is capable of escaping from over-smoothing issue. □

**Remark 7** (Stronger LFD). Based on the condition described in Proposition 3, when framelet is LFD, with sufficient small of $\mu$ or larger of $p$, it is not hard to verify that according to eq. (38), the p-Laplacian implicit propagation further shrink the Dirichlet energy of the node feature produced from framelet, and thus achieving a stronger LFD dynamic.

**Remark 8.** In Proposition 2 we showed that the Dirichlet energy of the node features produced from the implicit layer will not coverage to zero, indicating the robustness of the implicit layer in regarding to the over-smoothing issue. Additionally, we further verified in Lemma 3 that when graph framelet is with a monotonic dynamic (e.g., L/HFD), the inclusion of the implicit layer can even amplify the dynamic of framelet by a proper setting of $\mu$ and $p$. Our conclusion explicitly suggests the effectiveness on incorporating p-Laplacian based implicit propagation to multiscale GNNs which is with flexible control of model dynamics.

### 3.4 Proposed Model with Controlled Dynamics

Based on the aforementioned conclusions regarding energy behavior and the interaction between the implicit layer and framelet's energy dynamics, it becomes evident that irrespective of the homophily index of any given graph input, one can readily apply the condition of $\boldsymbol{\theta}(s)$ in Proposition 3 to facilitate the adaptation of the pL-UFG model to the input graph by simply adjusting the quantities of $\mu$ and $p$. This adjustment significantly reduces the training cost of the graph framelet. For instance, consider the case of employing a Haar type frame with $\ell = 1$, where we have only one low-pass and one high-pass domain. In this scenario, the trainable matrices for this model are $\boldsymbol{\theta}_{0,1}$, $\boldsymbol{\theta}_{1,1}$, and $\widehat{\mathbf{W}}$. Based on our conclusions, we can manually set both $\boldsymbol{\theta}_{0,1}$ and $\boldsymbol{\theta}_{1,1}$ to our requested quantities, thereby inducing either LFD or HFD. Consequently, the only remaining training cost is associated with $\widehat{\mathbf{W}}$, leading to large reduction on the overall training cost while preserving model's capability of handling both types of graphs. Accordingly, we proposed two additional pL-UFG variants with controlled model dynamics, namely **pL-UFG-LFD** and **pL-UFG-HFD**. More explicitly, the propagation of graph framelet with controlled dynamic takes the form as:

$$\mathbf{F}^{(k+1)} = \sigma\left(\mathcal{W}_{0,1}^{\top}\mathrm{diag}(\mathbf{1}_N)\mathcal{W}_{0,1}\mathbf{F}^{(k)}\widehat{\mathbf{W}} + \mathcal{W}_{1,1}^{\top}\mathrm{diag}(\theta\mathbf{1}_N)\mathcal{W}_{1,1}\mathbf{F}^{(k)}\widehat{\mathbf{W}}\right),$$

after which the output node features will be propagated through the iterative layers in defined in eq. (19) for the implicit layer eq. (18) for certain layers, and the resulting node feature will be forwarded to the next graph framelet convolution and implicit layer propagation. We note that to properly represent the Dirichlet energy of node features, we borrow the concept of electronic orbital energy levels in Figure. 1. The shaded outermost electrons correspond to higher energy levels, which can be analogously interpreted as higher variations in node features. Conversely, the closer the electrons are to the nucleus, the lower their energy levels, indicating lower variations in node features.

### 3.5 Equivalence to Non-Linear Diffusion

Diffusion on graph has gained its popularity recently (Chamberlain et al., 2021b; Thorpe et al., 2022) by providing a framework (i.e., PDE) to understand the GNNs architecture and as a principled way to develop a broad class of new methods. To the best of our knowledge, although the GNNs based on linear diffusion on graph (Chamberlain et al., 2021b;a; Thorpe et al., 2022) have been intensively explored, models built from non-linear graph diffusion have not attracted much attention in general. In this section, we aim to verify

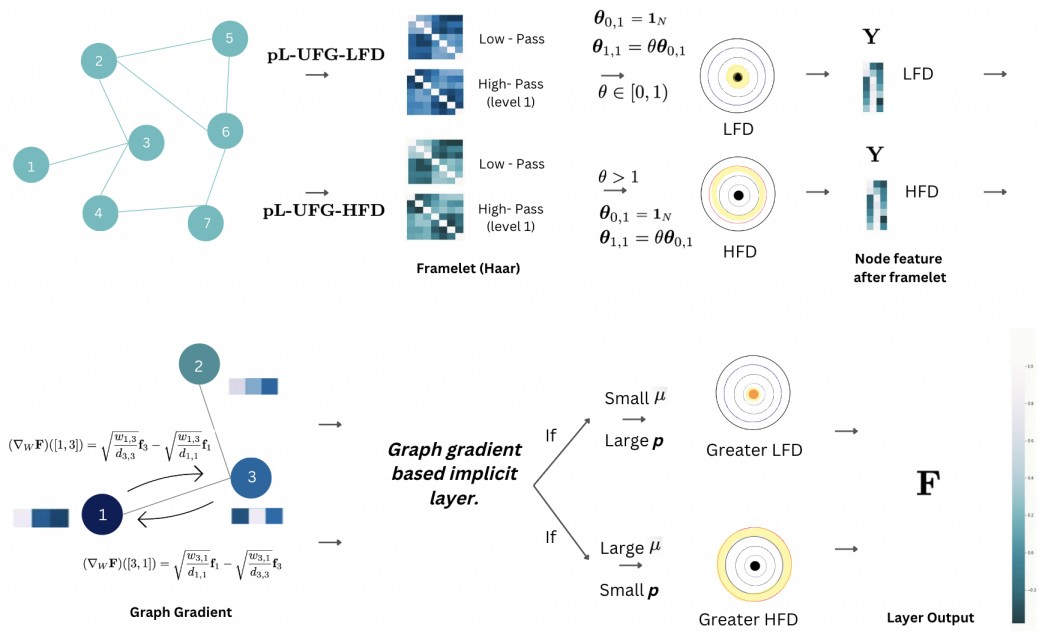

Figure 1: Illustration of the working flow of pL-UFG-LFD and pL-UFG-HFD under the Haar type frame with $\ell = 1$. The input graph features are first decomposed onto two frequency domains and further filtered by the diagonal matrix $\boldsymbol{\theta}_{0,1}$ and $\boldsymbol{\theta}_{1,1}$. With controlled model dynamics from Proposition 3 i.e., $\boldsymbol{\theta}_{0,1} = \mathbf{1}_N$ and $\boldsymbol{\theta}_{1,1} = \theta\boldsymbol{\theta}_{0,1}$, framelet can induce both LFD and HFD dynamics resulting as different level of Dirichlet energy of the produced node features. It is straightforward to check that when framelet is LFD, the level of node Dirichlet energy is less than its HFD counterpart. The generated node features from graph framelet is then inputted into p-Laplacian (with graph gradient as one component) based implicit layer. Based on our conclusions in Lemma 3 and Remark 7 with small/large quantity of $p$ and large/small quantity of $\mu$, the model's (framelet) dynamics are further strengthened resulting even smaller/higher energy levels.

that the iteration eq. (19) admits a scaled nonlinear diffusion with a source term. To see this, recall that p-Laplacian operator defined in eq. (14) has the form:

$$\Delta_p \mathbf{F} := -\frac{1}{2}\mathrm{div}(\|\nabla \mathbf{F}\|^{p-2}\nabla \mathbf{F}), \quad \text{for } p \geq 1. \tag{45}$$

Plugging in the definition of graph gradient and divergence defined in eq. (11) and eq. (13) into the above equation, one can compactly write out the form of p-Laplacian as:

$$(\Delta_p \mathbf{F})(i) = \sum_{v_j \sim v_i} \sqrt{\frac{w_{i,j}}{d_{i,i}}} \|\nabla_W \mathbf{F}([i,j])\|^{p-2} \left( \sqrt{\frac{w_{i,j}}{d_{i,i}}} \mathbf{f}_i - \sqrt{\frac{w_{i,j}}{d_{j,j}}} \mathbf{f}_j \right). \tag{46}$$

Furthermore, if we treat the iteration equation eq. (19) as a diffusion process, its forward Euler scheme has the form:

$$\begin{aligned} \frac{\mathbf{F}^{(k+1)} - \mathbf{F}^{(k)}}{\tau} &= \boldsymbol{\alpha}^{(k)}\mathbf{D}^{-1/2}\mathbf{M}^{(k)}\mathbf{D}^{-1/2}\mathbf{F}^{(k)} - \mathbf{F}^{(k)} + \boldsymbol{\beta}^{(k)}\mathbf{Y}, \\ &= \left( \boldsymbol{\alpha}^{(k)}\mathbf{D}^{-1/2}\mathbf{M}^{(k)}\mathbf{D}^{-1/2} - \mathbf{I} \right)\mathbf{F}^{(k)} + \boldsymbol{\beta}^{(k)}\mathbf{Y}. \end{aligned} \tag{47}$$

We set $\tau = 1$ for the rest of analysis for the convenience reasons. With all these setups, we summarize our results in the following:

**Lemma 4** (Non-Linear Diffusion)**.** *Assuming $\mathcal{G}$ is connected, the forward Euler scheme presented in eq. (47) admits a generalized non-linear diffusion on the graph. Specifically, we have:*

$$\left(\boldsymbol{\alpha}^{(k)}\mathbf{D}^{-1/2}\mathbf{M}^{(k)}\mathbf{D}^{-1/2} - \mathbf{I}\right)\mathbf{F}^{(k)} + \boldsymbol{\beta}^{(k)}\mathbf{Y} = \boldsymbol{\alpha}\left(\mathrm{div}(\|\nabla\mathbf{F}^{(k)}\|^{p-2}\nabla\mathbf{F}^{(k)})\right) + 2\mu\boldsymbol{\alpha}^{(k)}\mathbf{D}\mathbf{F}^{(k)} + 2\mu\boldsymbol{\alpha}^{(k)}\mathbf{F}^{(0)}.$$
(48)

*Proof.* The proof can be done by verification. We can explicitly write out the computation on the $i$-th row of the left hand side of eq. (48) as:

First let us denote the rows of $\mathbf{F}^{(k)}$ as $\mathbf{f}^{(k)}(i)$'s.

$$\sum_{v_j \sim v_i}\left(\alpha_{i,i}^{(k)}d_{ii}^{-1/2}M_{i,j}^{(k)}d_{jj}^{-1/2}\right)\mathbf{f}^{(k)}(j) - \mathbf{f}^{(k)}(i) + \beta_{i,i}^{(k)}Y(i)$$

$$=\alpha_{i,i}^{(k)}\left(\sum_{v_j \sim v_i}\left(\frac{M_{ij}}{\sqrt{d_{ii}}\sqrt{d_{jj}}}\mathbf{f}^{(k)}(j)\right) - \frac{1}{\alpha_{i,i}^{(k)}}\mathbf{f}^{(k)}(i)\right) + 2\mu\alpha_{i,i}^{(k)}\mathbf{f}^{(0)}(i)$$

$$=\alpha_{i,i}^{(k)}\left(\sum_{v_j \sim v_i}\left(\frac{M_{ij}}{\sqrt{d_{ii}}\sqrt{d_{jj}}}\mathbf{f}^{(k)}(j)\right) - \sum_{v_j \sim v_i}\left(\frac{M_{ij}}{d_{ii}} + 2\mu\right)\mathbf{f}^{(k)}(i)\right) + 2\mu\alpha_{i,i}^{(k)}\mathbf{f}^{(0)}(i)$$

$$=\alpha_{i,i}^{(k)}\left(\sum_{v_j \sim v_i}\sqrt{\frac{w_{i,j}}{d_{i,i}}}\|\nabla_W\mathbf{F}([i,j])\|^{p-2}\left(\sqrt{\frac{w_{i,j}}{d_{j,j}}}\mathbf{f}_j^{(k)} - \sqrt{\frac{w_{i,j}}{d_{i,i}}}\mathbf{f}_i^{(k)}\right) + 2\mu\sum_{v_j \sim v_i}\mathbf{f}_i^{(k)}\right)$$

$$+ 2\mu\alpha_{i,i}^{(k)}\mathbf{f}^{(0)}(i)$$

$$=\alpha_{i,i}^{(k)}\left((\Delta_p\mathbf{F})(i)\right) + 2\mu\alpha_{i,i}^{(k)}d_{ii}\mathbf{f}_i^{(k)} + 2\mu\alpha_{i,i}^{(k)}\mathbf{f}_i^{(0)}$$
(49)

When $i$ takes from 1 to $N$, it gives eq. (48) according to eq. (45) and eq. (46). Thus we complete the proof. $\square$

Based on the conclusion of Lemma 4, it is clear that the propagation via p-Laplacian implicit layer admits a scaled non-linear diffusion with two source terms. We note that the form of our non-linear diffusion coincidences to the one developed in (Chen et al., 2022b). However, in (Chen et al., 2022b) the linear operator is assigned via the calculation of graph Laplacian whereas in our model, the transformation acts over the whole p-Laplacian. Finally, it is worth noting that the conclusion in Lemma 4 can be transferred to the implicit schemes[8]. We omit it here.

**Remark 9.** With sufficiently large $\mu$ or small $p$, one can check that the strength of the diffusion, i.e. $\mathrm{div}(\|\nabla\mathbf{F}^{(k)}\|^{p-2}\nabla\mathbf{F}^{(k)})$, is diluted. Once two source terms $2\mu\boldsymbol{\alpha}^{(k)}\mathbf{D}\mathbf{F}^{(k)} + 2\mu\boldsymbol{\alpha}^{(k)}\mathbf{F}^{(0)}$ dominant the whole process, the generated node features approach to $\mathbf{D}\mathbf{F}^{(k)} + \mathbf{F}^{(0)}$, which suggests a framelet together with two source terms. The first term can be treated as the degree normalization of the node features from the last layer and the second term simply maintains the initial feature embedding. Therefore, the energy of the remaining node features in this case is just with the form presented in eq. (39), suggesting a preservation of node feature variations. Furthermore, this observation suggests our conclusion on the energy behavior of pL-UFG (Proposition 2); the interaction within pL-UFG described in Lemma 3 and Corollary 1 and lastly, the conclusion from Lemma 4 can be unified and eventually forms a well defined framework in assessing and understanding the property of pL-UFG.

## 4 Experiment

**Experiment outlines** In this section, we present comprehensive experimental results on the claims that we made from the theoretical aspects of our model. All experiments were conducted in PyTorch on NVIDIA Tesla V100 GPU with 5,120 CUDA cores and 16GB HBM2 mounted on an HPC cluster. In addition, for the sake of convenience, we listed the summary of each experimental section as follows:

---

[8]With a duplication of terminology, here the term "implicit" refers to the implicit scheme (i.e., backward propagation) in the training of the diffusion model.

- In Section 4.1, we show how a sufficient large/small $\mu$ can affect model's performance on heterophilic/homophilic graphs, and the results are almost invariant to the choice of $p$.

- In Section 4.2 we show some tests regarding to the results (i.e., Remark 7 and Lemma 3) of model's dynamics. Specifically, we verified the conclusions of stronger LFD and HFD in Section 3.3 with controlled model dynamics (quantity of $\theta$ ) of framelet to illustrate how the p-Laplacian based implicit layer interact with framelet model.

- In Section 4.3 we test the performances of pL-UFG-LFD and pL-UFG-HFD via real-world graph benchmarks versus various baseline models. Furthermore, as these two controllable pL-UFG models largely reduced the computational cost (as we claimed in Section 3.4), we show pL-UFG-LFD and pL-UFG-HFD can even handle the large-scale graph datasets and achieve remarkable learning accuracies.

**Hyper-parameter tuning** We applied exactly same hyper-parameter tunning strategy as (Shao et al., 2022) to make a fair comparsion. In terms of the settings for graph framelets, the framelet type is fixed as Haar ((Yang et al., 2022)) and the level $J$ is set to 1. The dilation scale $s \in \{1, 1.5, 2, 3, 6\}$, and for $n$, the degree of Chebyshev polynomial approximation is drawn from $\{2, 3, 7\}$. It is worth noting that in graph framelets, the Chebyshev polynomial is utilized for approximating the spectral filtering of the Laplacian eigenvalues. Thus, a $d$-degree polynomial approximates $d$-hop neighbouring information of each node of the graph. Therefore, when the input graph is heterophilic, one may prefer to have a relatively larger $d$ as node labels tend to be different between directly connected (1-hop) nodes.

### 4.1 Synthetic Experiment on Variation of $\mu$

**Setup** In this section, we show how a sufficiently large/small of $\mu$ can affect model's performance on hetero/homophilic graphs. In order to make a fair comparison, all the parameters of pL-UFG followed the settings included in (Shao et al., 2022). For this test, we selected two datasets: `Cora` (heterophilic index: 0.825, 2708 nodes and 5278 edges) and `Wisconsin` (heterophilic index: 0.15, 499 nodes and 1703 edges) from `https://www.pyg.org/`. We assigned the quantity of $p = \{1, 1.5, 2, 2.5\}$ combined with a set of $\mu = \{0.1, 0.5, 1, 5, 10, 20, 30, 50, 70\}$. The number of epochs was set to 200 and the test accuracy (in %) is obtained as the average test accuracy of 10 runs.

**Results and Discussion** The experimental results are presented in Figure 2. When examining the results obtained through the homophily graph (Figure 2a), it is apparent that all variants of pL-UFGs achieved the best performance when $\mu = 0.1$, which is the minimum value of $\mu$. As the value of $\mu$ increased, the learning accuracy decreased. This suggests that a larger sharpening effect was induced by the model, as stated in Remark 7 and Proposition 2, causing pL-UFGs to incorporate higher amounts of Dirichlet energy into their generated node features. Consequently, pL-UFGs are better suited for adapting to heterophily graphs. This observation is further supported by the results in Figure 2b, where all pL-UFG variants achieved their optimal performance with a sufficiently large $\mu$ when the input graph is heterophilic.

Additional interesting observation on the above result is despite the fact that all model variants demonstrated superior learning outcomes on both homophilic and heterophilic graphs when assigned sufficiently large or small values of $\mu$, it can be observed that when the quantity of $p$ is small, pL-UFG requires a smaller value of $\mu$ to fit the heterophily graph (blue line in Fig. 2b). On the other hand, when the models have relatively large value of $p$ (i.e., $p = 2.5$), it is obvious that these models yielded the most robust results when there is an increase of $\mu$ (red line in Fig. 2a). These phenomena further support the notion that $p$ and $\mu$ exhibit opposite effects on the model's energy behavior as well as its adaptation to homophily and heterophily graphs.

### 4.2 Synthetic Experiment on Testing of Model's Dynamics

Now, we take one step ahead. Based on Lemma 3 and Remark 7, with the settings of $\theta$ provided in Proposition 3, the inclusion of $p$-Laplacian based implicit layer can further enhance framelet's LFD and HFD dynamics. This suggests that one can control the entries of $\theta$ based on the conditions provided in Proposition 3 and by

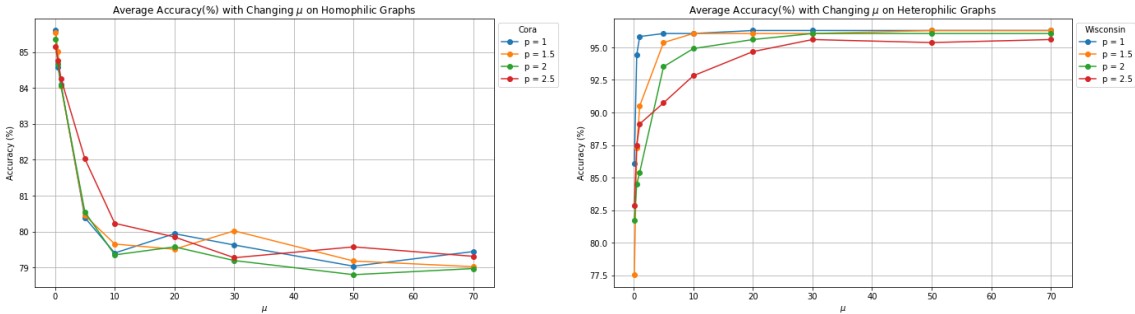

(a) Accuracy on `Cora` via different combinations of $\mu$ and $p$.

(b) Accuracy on `Wisconsin` via different combinations of $\mu$ and $p$.

Figure 2: Performance of pL-UFG with various combinations of the values of $\mu$ and $p$.

only changing the quantity of $\mu$ and $p$ to test model's adaption power on both homophily and heterophily graphs. Therefore, in this section, we show how a (dynamic) controlled framelet model can be further enhanced by the assistant from the $p$-Laplacian regularizer. Similarly, we applied the same setting to the experiments in (Shao et al., 2022).

**Setup and Results** To verify the claims on in Lemma 3 and Remark 7, we deployed the same settings mentioned in Proposition 3. Specifically, we utilized Haar frame with $\ell = 1$ and set $\boldsymbol{\theta}_{0,1} = \mathbf{I}_N$, $\boldsymbol{\theta}_{0,1} = \theta \mathbf{I}_N$. For heterophilic graphs (`Wisconsin`), $\theta = 2$, and for the homophilic graph (`Cora`), $\theta = 0.2$. The result of the experiment is presented in Figure 3. Similar to the results observed from Section 4.1, it is shown that when the relatively large quantity of $\mu$ is assigned, model's capability of adapting to homophily/heterophily graph decreased/increased. This directly verifies that the p-Laplacian based implicit layer interacts and further enhances the (controlled) dynamic of the framelet by the value of $p$ and $\mu$, in terms of adaptation.

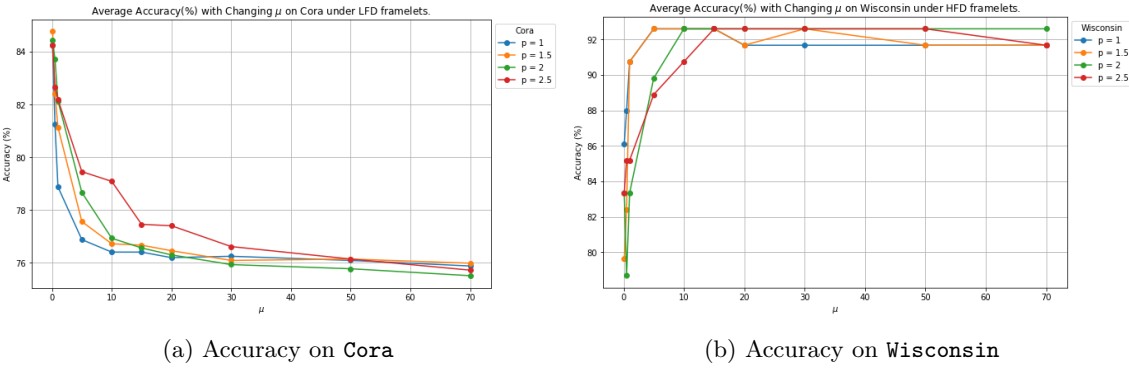

(a) Accuracy on `Cora`

(b) Accuracy on `Wisconsin`

Figure 3: Average Accuracy(%) with Changing $\mu$ and $p$ under (manually fixed) LFD/HFD framelet models. All framelet model in Fig. 3a are LFD dynamic with $\theta_{0,1} = \mathbf{I}_N$, $\theta_{1,1} = \theta \mathbf{1}_N$, $\theta = 0.2$. On Fig. 3b, all framelet models are HFD with $\theta_{0,1} = \mathbf{I}_N$, $\theta_{1,1} = \theta \mathbf{1}_N$, $\theta = 2$.

### 4.3 Real-world Node Classification and Scalability

Previous synthetic numerical results show predictable performance of both pL-UFG-LFD and pL-UFG-HFD. In this section, we present the learning accuracy of our proposed models via real-world homophily and heterophily graphs. Similarly, we deployed the same experimental setting from Shao et al. (2022). In addition, to verify the claim in Remark 3.4, we tested our proposed model via large-scale graph dataset (`ogbn-arxiv`) to show the proposed model's scalability which is rarely explored. We include the summary statistic of the datasets in Table 2. All datasets are split according to Hamilton et al. (2017).

For the settings of $\mu$, $p$ and $\theta$ within pL-UFG-LFD and pL-UFG-HFD, we assigned $\mu = \{0.1, 0.5, 1, 2.0\}$, $p = \{1, 1.5, 2, 2.5\}$ and $\theta = \{0.2, 0.5, 0.8\}$ for pL-UFG-LFD in order to fit the homophily graphs, and for pL-UFG-HFD, we assigned $\mu = \{10, 20, 30\}$, $p = \{1, 1.5, 2, 2.0, 2.5\}$ and $\theta = \{5, 7.5, 10\}$ for heterophily graphs. The learning accuracy are presented in Table 3 and 4. Furthermore, rather than only reporting the average accuracy and related standard deviation, to further verify the significance of the improvement, we also computed the 95% confidence interval under $t$-distribution for the highest learning accuracy of the baselines and mark $*$ to our model's learning accuracy if it is outside the confidence interval.

We include a brief introduction on the baseline models used in this experiment:

- **MLP**: Standard feedward multiple layer perceptron.

- **GCN** Kipf & Welling (2016): GCN is the first of its kind to implement linear approximation to spectral graph convolutions.

- **SGC** Wu et al. (2019b): SGC reduces GCNs' complexity by removing nonlinearities and collapsing weight matrices between consecutive layers. Thus serves as a more powerful and efficient GNN baseline.

- **GAT** Veličković et al. (2018): GAT generates attention coefficient matrix that element-wisely multiplied on the graph adjacency matrix according to the node feature based attention mechanism via each layer to propagate node features via the relative importance between them.

- **JKNet** Xu et al. (2018): JKNet offers the capability to adaptively exploit diverse neighbourhood ranges, facilitating enhanced structure-aware representation for individual nodes.

- **APPNP** Gasteiger et al. (2019): APPNP leverages personalized PageRank to disentangle the neural network from the propagation scheme, thereby merging GNN functionality.

- **GPRGNN** Chien et al. (2021): The GPRGNN architecture dynamically learns General Pagerank (GPR) weights to optimize the extraction of node features and topological information from a graph, irrespective of the level of homophily present.

- **p-GNN** Fu et al. (2022a):p-GNN is a $p$-Laplacian based graph neural network model that incorporates a message-passing mechanism derived from a discrete regularization framework. To make a fair comparison, we test p-GNN model with different quantity of $p$.

- **UFG** Zheng et al. (2022b): UFG, a class of GNNs built upon framelet transforms utilizes framelet decomposition to effectively merge graph features into low-pass and high-pass spectra.

- **pL-UFG** Shao et al. (2022): pL-UFG employs a p-Laplacian based implicit layer to enhance the adaptability of multi-scale graph convolution networks (i.e.,UFG) to filter-based domains, effectively improving the model's adaptation to both homophily and heterophily graphs. Furthermore, as two types of pL-UFG models are proposed in Shao et al. (2022), we test both two pL-UFG variants as our baseline models. For more details including the precise formulation of the model, please check Shao et al. (2022).

**Discussion on the Results, Scalability and Computational Complexity** From both Table 3 and 4, it is clear that our proposed model (pL-UFG-LFD and pL-UFG-HFD) produce state-of-the-art learning accuracy compared to various baseline models. For the datasets (i.e.,`Pubmed` and `Squirrel`) on which pL-UFG-LFD and pL-UFG-HFD are not the best, one can observe that pL-UFG-LFD and pL-UFG-HFD still have nearly identical learning outcomes compared to the best pL-UFG results. This suggests even within the pL-UFG with controlled framelet dynamics, by adjusting the values of $\mu$ and $p$, our proposed models are still able to generate state-of-the-art learning results with the computational complexity largely reduced compared to the pL-UFG and UFG. This observation directly verifies Lemma 3 and Remark 7. In addition, due to the reduction of computational cost, our dynamic controlled models (pL-UFG-LFD and pL-UFG-HFD) show a strong capability of handling the large-scale graph dataset, which is a challenging issue (scalability) for some

Table 2: Statistics of the datasets, $\mathcal{H}(\mathcal{G})$ represent the level of homophily of overall benchmark datasets.

| Datasets | Class | Feature | Node | Edge | $\mathcal{H}(\mathcal{G})$ |
|---|---|---|---|---|---|
| Cora | 7 | 1433 | 2708 | 5278 | 0.825 |
| CiteSeer | 6 | 3703 | 3327 | 4552 | 0.717 |
| PubMed | 3 | 500 | 19717 | 44324 | 0.792 |
| Computers | 10 | 767 | 13381 | 245778 | 0.802 |
| Photo | 8 | 745 | 7487 | 119043 | 0.849 |
| CS | 15 | 6805 | 18333 | 81894 | 0.832 |
| Physics | 5 | 8415 | 34493 | 247962 | 0.915 |
| Arxiv | 23 | 128 | 169343 | 1166243 | 0.681 |
| Chameleon | 5 | 2325 | 2277 | 31371 | 0.247 |
| Squirrel | 5 | 2089 | 5201 | 198353 | 0.216 |
| Actor | 5 | 932 | 7600 | 26659 | 0.221 |
| Wisconsin | 5 | 251 | 499 | 1703 | 0.150 |
| Texas | 5 | 1703 | 183 | 279 | 0.097 |
| Cornell | 5 | 1703 | 183 | 277 | 0.386 |

GNNs especially multi-scale graph convolutions such as framelets (Zheng et al., 2022b) without additional data pre-processing steps. Accordingly, one can check that pL-UFG-LFD outperforms all included baselines on `Arxiv` datasets. Lastly, one can also find that the most of the improvements between the learning accuracy produced from our model and the baselines are significant.

### 4.4 Limitation of the Proposed Models and Future Studies

First, we note that our analysis on the convergence, energy dynamic and equivalence between our proposed model can be applied or partially applied to most of existing GNNs. Based on we have claimed in regarding to the theoretical perspective of pL-UFG, although we assessed model property via different perspective, eventually all theoretical conclusions come to the same conclusion (i.e., the asymptotic behavior of pL-UFG). Therefore, it would be beneficial to deploy our analyzing framework to other famous GNNs. Since the main propose of this paper is to re-assess the property of pL-UFG, we leave this to the future work.

In addition, to induce LFD/HFD to pL-UFG, we set the value of $\theta$ as constant according to Proposition 3, however, due to large variety of real-world graphs, it is challenging to determine the most suitable $\theta$ when we fix it as a constant. This suggests the exploration on controlling model's dynamic via selecting $\theta$ is still rough. Moreover, based on Definition 1, the homophily index of a graph is summary statistic over all nodes. However, even in the highly homophilic graph, there are still some nodes with their neighbours with different labels. This suggests the index is only capable of presenting the global rather than local labelling information of the graph. Accordingly, assigning a constant $\theta$ to induce LFD/HFD might not be able to equip pL-UFG enough power to capture detailed labelling information of the graph. Therefore, another future research direction is to potentially explore the design of $\theta$ via the local labelling information of the graph. Finally, we note that another consequence of setting $\theta_{0,1}$ and $\theta_{1,1}$ as constant is such setting narrows the model's parameter space, as one can check the only learnable matrix left via explicit part of pL-UFG (eq. (9)) is $\widehat{\mathbf{W}}$. Accordingly, the narrowed parameter space might make the solution of the model optimization apart from desired solution as before, causing potential increase of learning variance.

## 5 Concluding Remarks

In this work, we performed theoretical analysis on pL-UFG. Specifically, we verified that by choosing suitable quantify of the model parameters ($\mu$ and $p$), the implicit propagation induced from p-Laplacian is capable of amplifying or shrinking the Dirichlet energy of the node features produced from the framelet. Consequently, such manipulation of the energy results in a stronger energy dynamic of framelet and therefore enhancing model's adaption power on both homophilic and heterophilic graphs. We further explicitly showed the proof

Table 3: Test accuracy (%) on homophilic graphs, the top learning accuracy is highlighted in **bold** and the second accuracy is underlined. The term OOM means out of memory.

| Method | Cora | CiteSeer | PubMed | Computers | Photos | CS | Physics | Arxiv |
|---|---|---|---|---|---|---|---|---|
| MLP | 66.04±1.11 | 68.99±0.48 | 82.03±0.24 | 71.89±5.36 | 86.11±1.35 | 93.50±0.24 | 94.56±0.11 | 55.50±0.78 |
| GCN | 84.72±0.38 | 75.04±1.46 | 83.19±0.13 | 78.82±1.87 | 90.00±1.49 | 93.00±0.12 | 95.55±0.09 | 70.07±0.79 |
| SGC | 83.79±0.37 | 73.52±0.89 | 75.92±0.26 | 77.56±0.88 | 86.44±0.35 | 92.18±0.22 | 94.99±0.13 | 71.01±0.30 |
| GAT | 84.37±1.13 | 74.80±1.00 | 83.92±0.28 | 78.68±2.09 | 89.63±1.75 | 92.57 ±0.14 | 95.13±0.15 | OOM |
| JKNet | 83.69±0.71 | 74.49±0.74 | 82.59±0.54 | 69.32±3.94 | 86.12±1.12 | 91.11±0.22 | 94.45±0.33 | OOM |
| APPNP | 83.69±0.71 | 75.84±0.64 | 80.42±0.29 | 73.73±2.49 | 87.03±0.95 | 91.52±0.14 | 94.71±0.11 | OOM |
| GPRGNN | 83.79±0.93 | 75.94±0.65 | 82.32±0.25 | 74.26±2.94 | 88.69±1.32 | 91.89 ±0.08 | 94.85±0.23 | OOM |
| UFG | 80.64±0.74 | 73.30±0.19 | 81.52±0.80 | 66.39±6.09 | 86.60±4.69 | 95.27±0.04 | 95.77±0.04 | 71.08±0.49 |
| PGNN$^{1.0}$ | 84.21±0.91 | 75.38±0.82 | 84.34±0.33 | 81.22±2.62 | 87.64±5.05 | 94.88±0.12 | 96.15±0.12 | OOM |
| PGNN$^{1.5}$ | 84.42±0.71 | 75.44±0.98 | 84.48±0.21 | 82.68±1.15 | 91.83±0.77 | 94.13±0.08 | 96.14±0.08 | OOM |
| PGNN$^{2.0}$ | 84.74±0.67 | 75.62±1.07 | 84.25 ±0.35 | 83.40±0.68 | 91.71±0.93 | 94.28±0.10 | 96.03±0.07 | OOM |
| PGNN$^{2.5}$ | 84.48±0.77 | 75.22±0.73 | 83.94±0.47 | 82.91±1.34 | 91.41±0.66 | 93.40±0.07 | 95.75±0.05 | OOM |
| pL-UFG1$^{1.0}$ | 84.54±0.62 | 75.88±0.60 | 85.56±0.18 | 82.07±2.78 | 85.57±19.92 | 95.03±0.22 | 96.19±0.06 | 70.28±9.13 |
| pL-UFG1$^{1.5}$ | 84.96±0.38 | 76.04±0.85 | 85.59±0.18 | 85.04±1.06 | 92.92±0.37 | 95.03±0.22 | 96.27±0.06 | 71.25±8.37 |
| pL-UFG1$^{2.0}$ | 85.20±0.42 | 76.12±0.82 | 85.59±0.17 | 85.26±1.15 | 92.65±0.65 | 94.77±0.27 | 96.04±0.07 | OOM |
| pL-UFG1$^{2.5}$ | 85.30±0.60 | 76.11±0.82 | 85.54±0.18 | 85.18±0.88 | 91.49±1.29 | 94.86±0.14 | 95.96±0.11 | OOM |
| pL-UFG2$^{1.0}$ | 84.42±0.32 | 74.79± 0.62 | 85.45±0.18 | 84.88±0.84 | 85.30±19.50 | 95.03±0.19 | 96.06±0.11 | 71.01±7.28 |
| pL-UFG2$^{1.5}$ | 85.60±0.36 | 75.61±0.60 | 85.59±0.18 | 84.55±1.57 | 93.00±0.61 | 95.03±0.19 | 96.14±0.09 | 71.21±6.19 |
| pL-UFG2$^{2.0}$ | 85.20±0.42 | 76.12±0.82 | **85.59±0.17** | 85.27±1.15 | 92.50±0.40 | 94.77±0.27 | 96.05±0.07 | OOM |
| pL-UFG-LFD | **85.64±1.36** | **77.39*±1.59** | 85.08±1.33 | **85.36*±1.39** | **93.17*±1.30** | **96.13*±1.08** | **96.49*±1.04** | **71.96±1.25** |

of the convergence of pL-UFG, which to our best of knowledge, fills the knowledge gap at least in the field of p-Laplacian based multi-scale GNNs. Moreover, we showed the equivalence between pL-UFG and the non-linear graph diffusion, indicating that pL-UFG can be trained via various training schemes. Finally, it should be noted that for the simplicity of the analysis, we have made several assumptions and only focus on the Haar type frames. It suffices in regards to the scope of this work. However, it would be interesting to consider more complex energy dynamics by reasonably dropping some of the assumptions or from other types of frames, we leave this to future work.

# References

Wendong Bi, Lun Du, Qiang Fu, Yanlin Wang, Shi Han, and Dongmei Zhang. Make heterophily graphs better fit gnn: A graph rewiring approach, 2022.

Table 4: Test accuracy (%) on heterophilic graphs. the top learning accuracy is highlighted in **bold** and the second accuracy is underlined.

| Method | Chameleon | Squirrel | Actor | Wisconsin | Texas | Cornell |
|---|---|---|---|---|---|---|
| MLP | 48.82±1.43 | 34.30±1.13 | 41.66±0.83 | 93.45±2.09 | 71.25±12.99 | 83.33±4.55 |
| GCN | 33.71±2.27 | 26.19±1.34 | 33.46±1.42 | 67.90±8.16 | 53.44±11.23 | 55.68±10.57 |
| SGC | 33.83±1.69 | 26.89±0.98 | 32.08±2.22 | 59.56±11.19 | 64.38±7.53 | 43.18±16.41 |
| GAT | 41.95±2.65 | 25.66±1.72 | 33.64±3.45 | 60.65±11.08 | 50.63±28.36 | 34.09±29.15 |
| JKNet | 33.50±3.46 | 26.95±1.29 | 31.14±3.63 | 60.42±8.70 | 63.75±5.38 | 45.45±9.99 |
| APPNP | 34.61±3.15 | 32.61±0.93 | 39.11±1.11 | 82.41±2.17 | 80.00±5.38 | 60.98±13.44 |
| GPRGNN | 34.23±4.09 | 34.01±0.82 | 34.63±0.58 | 86.11±1.31 | 84.38±11.20 | 66.29±11.20 |
| UFG | 50.11±1.67 | 31.48±2.05 | 40.13±1.11 | 93.52±2.36 | 84.69±4.87 | 83.71±3.28 |
| PGNN$^{1.0}$ | 49.04±1.16 | 34.79±1.01 | 40.91±1.41 | 94.35±2.16 | 82.00±11.31 | 82.73±6.92 |
| PGNN$^{1.5}$ | 49.12±1.14 | 34.86±1.25 | 40.87±1.47 | 94.72±1.91 | 81.50±10.70 | 81.97±10.16 |
| PGNN$^{2.0}$ | 49.34±1.15 | 34.97±1.41 | 40.83±1.81 | 94.44±1.75 | 84.38±11.52 | 81.06±10.18 |
| PGNN$^{2.5}$ | 49.16±1.40 | 34.94±1.57 | 40.78±1.51 | 94.35±2.16 | 83.38±12.95 | 81.82±8.86 |
| pL-UFG1$^{1.0}$ | 56.81±1.69 | 38.81±1.97 | 41.26±1.66 | 96.48±0.94 | 86.13±7.47 | 86.06±3.16 |
| pL-UFG1$^{1.5}$ | 56.89±1.17 | 39.73±1.22 | 40.95±0.93 | 96.48±1.07 | 87.00±5.16 | 86.52±2.29 |
| pL-UFG1$^{2.0}$ | 56.24±1.02 | 39.72±1.86 | 40.95±0.93 | 96.59±0.72 | 86.50±8.84 | 85.30±2.35 |
| pL-UFG1$^{2.5}$ | 56.11±1.25 | 39.38±1.78 | 41.04±0.99 | 95.34±1.64 | 89.00±4.99 | 83.94±3.53 |
| pL-UFG2$^{1.0}$ | 55.51±1.53 | 36.94±5.69 | 29.28±19.25 | 93.98±2.94 | 85.00±5.27 | 87.73±2.49 |
| pL-UFG2$^{1.5}$ | 57.22±1.19 | **39.80±1.42** | 40.89±0.75 | 96.48±0.94 | 87.63±5.32 | 86.82±1.67 |
| pL-UFG2$^{2.0}$ | 56.19±0.99 | 39.74±1.66 | 41.01±0.80 | 96.14±1.16 | 86.50±8.84 | 85.30±2.35 |
| pL-UFG2$^{2.5}$ | 55.69±1.15 | 39.30±1.68 | 40.86±0.74 | 95.80±1.44 | 86.38±2.98 | 84.55±3.31 |
| pL-fUFG$^{1.0}$ | 55.80±1.93 | 38.43±1.26 | 32.84±16.54 | 93.98±3.47 | 86.25±6.89 | 87.27±2.27 |
| pL-fUFG$^{1.5}$ | 55.65±1.96 | 38.40±1.52 | 41.00±0.99 | 96.48±1.29 | 87.25±3.61 | 86.21±2.19 |
| pL-fUFG$^{2.0}$ | 55.95±1.29 | 38.33±1.71 | 41.25±0.84 | 96.25±1.25 | 88.75±4.97 | 83.94±3.78 |
| pL-fUFG$^{2.5}$ | 55.56±1.66 | 38.39±1.48 | 40.55±0.50 | 95.28±2.24 | 88.50±7.37 | 83.64±3.88 |
| pL-UFG-HFD | **58.60$^*$±1.74** | 39.63±2.01 | **44.63$^*$±2.75** | **96.64±1.77** | **89.31±8.40** | **88.97$^*$±3.36** |

Cristian Bodnar, Francesco Di Giovanni, Benjamin Paul Chamberlain, Pietro Liò, and Michael M Bronstein. Neural sheaf diffusion: A topological perspective on heterophily and oversmoothing in gnns. *arXiv:2202.04579*, 2022.

Michael M Bronstein, Joan Bruna, Taco Cohen, and Petar Veličković. Geometric deep learning: Grids, groups, graphs, geodesics, and gauges. *arXiv preprint arXiv:2104.13478*, 2021.

Ben Chamberlain, James Rowbottom, Maria I Gorinova, Michael Bronstein, Stefan Webb, and Emanuele Rossi. Grand: Graph neural diffusion. In *International Conference on Machine Learning*, pp. 1407–1418.

PMLR, 2021a.

Benjamin Chamberlain, James Rowbottom, Davide Eynard, Francesco Di Giovanni, Xiaowen Dong, and Michael Bronstein. Beltrami flow and neural diffusion on graphs. In M. Ranzato, A. Beygelzimer, Y. Dauphin, P.S. Liang, and J. Wortman Vaughan (eds.), *Advances in Neural Information Processing Systems*, volume 34, pp. 1594–1609. Curran Associates, Inc., 2021b. URL https://proceedings.neurips.cc/paper_files/paper/2021/file/0cbed40c0d920b94126eaf5e707be1f5-Paper.pdf.

Jialin Chen, Yuelin Wang, Cristian Bodnar, Pietro Liò, and Yu Guang Wang. Dirichlet energy enhancement of graph neural networks by framelet augmentation. *github*, 2022a.

Qi Chen, Yifei Wang, Yisen Wang, Jiansheng Yang, and Zhouchen Lin. Optimization-induced graph implicit nonlinear diffusion. In *International Conference on Machine Learning*, pp. 3648–3661. PMLR, 2022b.

Eli Chien, Jianhao Peng, Pan Li, and Olgica Milenkovic. Adaptive universal generalized pagerank graph neural network. In *Proceedings of International Conference on Learning Representations*, 2021.

Fan RK Chung. *Spectral graph theory*, volume 92. American Mathematical Soc., 1997.

Francesco Di Giovanni, James Rowbottom, Benjamin P Chamberlain, Thomas Markovich, and Michael M Bronstein. Graph neural networks as gradient flows. *arXiv:2206.10991*, 2022.

Bin Dong. Sparse representation on graphs by tight wavelet frames and applications. *Applied and Computational Harmonic Analysis*, 42(3):452–479, 2017. doi: 10.1016/j.acha.2015.09.005.

Pavel Drábek and Stanislav I Pohozaev. Positive solutions for the p-laplacian: application of the fibrering method. *Proceedings of the Royal Society of Edinburgh Section A: Mathematics*, 127(4):703–726, 1997.

Guoji Fu, Peilin Zhao, and Yatao Bian. *p*-Laplacian based graph neural networks. In *Proceedings of the 39th International Conference on Machine Learning*, volume 162 of *PMLR*, pp. 6878–6917, 2022a.

Guoji Fu, Peilin Zhao, and Yatao Bian. *p*-laplacian based graph neural networks. In *International Conference on Machine Learning*, pp. 6878–6917. PMLR, 2022b.

JP García Azorero and I Peral Alonso. Existence and nonuniqueness for the p-laplacian. *Communications in Partial Differential Equations*, 12(12):126–202, 1987.

Johannes Gasteiger, Aleksandar Bojchevski, and Stephan Günnemann. Predict then propagate: Graph neural networks meet personalized pagerank. In *Proceedings of International Conference on Learning Representations*, 2019.

Will Hamilton, Zhitao Ying, and Jure Leskovec. Inductive representation learning on large graphs. *Advances in Neural Information Processing Systems*, 30, 2017.

David K Hammond, Pierre Vandergheynst, and Rémi Gribonval. Wavelets on graphs via spectral graph theory. *Applied and Computational Harmonic Analysis*, 30(2):129–150, 2011.

Andi Han, Dai Shi, Zhiqi Shao, and Junbin Gao. Generalized energy and gradient flow via graph framelets. *arXiv preprint arXiv:2210.04124*, 2022.

Bernd Kawohl and Jiri Horak. On the geometry of the *p*-laplacian operator. *arXiv preprint arXiv:1604.07675*, 2016.

Thomas N Kipf and Max Welling. Semi-supervised classification with graph convolutional networks. *arXiv preprint arXiv:1609.02907*, 2016.

Qimai Li, Zhichao Han, and Xiao-Ming Wu. Deeper insights into graph convolutional networks for semi-supervised learning. In *AAAI Conference on Artificial Intelligence*, 2018.

Remigijus Paulavičius and Julius Žilinskas. Analysis of different norms and corresponding lipschitz constants for global optimization. *Technological and Economic Development of Economy*, 12(4):301–306, 2006.

Hongbin Pei, Bingzhe Wei, Kevin Chen-Chuan Chang, Yu Lei, and Bo Yang. Geom-GCN: Geometric graph convolutional networks. In *International Conference on Learning Representations*, 2019.

Zhiqi Shao, Andi Han, Dai Shi, Andrey Vasnev, and Junbin Gao. Generalized Laplacian regularized framelet gcns. *arXiv:2210.15092*, 2022.

Dai Shi, Yi Guo, Zhiqi Shao, and Junbin Gao. How curvature enhance the adaptation power of framelet gcns. *arXiv preprint arXiv:2307.09768*, 2023.

Wim Sweldens. The lifting scheme: A construction of second generation wavelets. *SIAM Journal on Mathematical Analysis*, 29(2):511–546, 1998. doi: 10.1137/S0036141095289051.

Matthew Thorpe, Tan Minh Nguyen, Hedi Xia, Thomas Strohmer, Andrea Bertozzi, Stanley Osher, and Bao Wang. GRAND++: Graph neural diffusion with a source term. In *International Conference on Learning Representations*, 2022. URL https://openreview.net/forum?id=EMxu-dzvJk.

César Torres. Boundary value problem with fractional p-laplacian operator. *arXiv preprint arXiv:1412.6438*, 2014.

Petar Veličković, Guillem Cucurull, Arantxa Casanova, Adriana Romero, Pietro Liò, and Yoshua Bengio. Graph attention networks. In *International Conference on Learning Representations*, 2018.

Yifei Wang, Yisen Wang, Jiansheng Yang, and Zhouchen Lin. Dissecting the diffusion process in linear graph convolutional networks. *Advances in Neural Information Processing Systems*, 34:5758–5769, 2021.

Quanmin Wei, Jinyan Wang, Jun Hu, Xianxian Li, and Tong Yi. Ogt: optimize graph then training gnns for node classification. *Neural Computing and Applications*, 34(24):22209–22222, 2022. doi: 10.1007/s00521-022-07677-5. URL https://doi.org/10.1007/s00521-022-07677-5.

Felix Wu, Amauri Souza, Tianyi Zhang, Christopher Fifty, Tao Yu, and Kilian Weinberger. Simplifying graph convolutional networks. In *International Conference on Machine Learning*, pp. 6861–6871. PMLR, 2019a.

Felix Wu, Tianyi Zhang, Amauri Holanda de Souza, Christopher Fifty, Tao Yu, and Kilian Q. Weinberger. Simplifying graph convolutional networks. In *Proceedings of International Conference on Machine Learning*, 2019b.

Zonghan Wu, Shirui Pan, Fengwen Chen, Guodong Long, Chengqi Zhang, and S Yu Philip. A comprehensive survey on graph neural networks. *IEEE transactions on Neural Networks and Learning Systems*, 32(1):4–24, 2020.

Keyulu Xu, Chengtao Li, Yonglong Tian, Tomohiro Sonobe, Ken-ichi Kawarabayashi, and Stefanie Jegelka. Representation learning on graphs with jumping knowledge networks. In *Proceedings of International Conference on Machine Learning*, 2018.

Keyulu Xu, Weihua Hu, Jure Leskovec, and Stefanie Jegelka. How powerful are graph neural networks? In *International Conference on Learning Representations*, 2019. URL https://openreview.net/forum?id=ryGs6iA5Km.

Mengxi Yang, Xuebin Zheng, Jie Yin, and Junbin Gao. Quasi-framelets: Another improvement to graph neural networks. *arXiv:2201.04728*, 2022.

Xin Zheng, Yixin Liu, Shirui Pan, Miao Zhang, Di Jin, and Philip S. Yu. Graph neural networks for graphs with heterophily: A survey, 2022a.

Xuebin Zheng, Bingxin Zhou, Junbin Gao, Yuguang Wang, Pietro Lió, Ming Li, and Guido Montufar. How framelets enhance graph neural networks. In *International Conference on Machine Learning*, pp. 12761–12771. PMLR, 2021.

Xuebin Zheng, Bingxin Zhou, Yu Guang Wang, and Xiaosheng Zhuang. Decimated framelet system on graphs and fast g-framelet transforms. *Journal of Machine Learning Research*, 23:18–1, 2022b.

Bingxin Zhou, Ruikun Li, Xuebin Zheng, Yu Guang Wang, and Junbin Gao. Graph denoising with framelet regularizer. *IEEE Transactions on Artificial Intelligence*, 2021.

Dengyong Zhou and Bernhard Schölkopf. Regularization on discrete spaces. In *Joint Pattern Recognition Symposium*, pp. 361–368. Springer, 2005.

Jiong Zhu, Yujun Yan, Lingxiao Zhao, Mark Heimann, Leman Akoglu, and Danai Koutra. Beyond homophily in graph neural networks: Current limitations and effective designs. In *Advances in Neural Information Processing Systems*, volume 33, pp. 7793–7804, 2020.

Meiqi Zhu, Xiao Wang, Chuan Shi, Houye Ji, and Peng Cui. Interpreting and unifying graph neural networks with an optimization framework. In *Proceedings of the Web Conference 2021*, pp. 1215–1226, 2021.

Chunya Zou, Andi Han, Lequan Lin, and Junbin Gao. A simple yet effective SVD-GCN for directed graphs. *arXiv:2205.09335*, 2022.

