# OpenReview forum: "Revisiting Generalized p-Laplacian Regularized Framelet GCNs: Convergence, Energy Dynamic and as Non-Linear Diffusion"
_TMLR — Rejected by TMLR_

### Review · Reviewer_me6p · 2023-09-07

**Summary Of Contributions:**

The paper performs a theoretical analysis of "Generalized p-Laplacian Regularized Framelet GCNs", (pL-UFG where the U stands for undecimated) a method combining chebyshev polynomial approximation of the graph laplacian with wavelet transforms and $p$  regularization which attempts to create a spatially localized alternative to spectral GCN, introduced in https://arxiv.org/abs/2210.15092 and prove in particular

- that the iterative procedure which is used to perform the regularization does indeed provably converge with suitable choice of regularization weight $\mu$
- that by increasing/shrinking $\mu$ the network can be biased  to increase/decrease the dirichlet energy of the node embeddings and can avoid oversmoothing even in the asymptote
- that by using Haar-type filters and specific settings of the parameters, one can bias the network towards low or high frequency dominance, tuning the model towards homophilic (LFD) or heterophilic (HFD) datasets
- use this manual setting to create a variant of the pL-UFG with reduced training load, scaling to previously prohibitvely large datasets
- establish that the regularization is equivalent to a non-linear diffusion process

And then performs empirical validation of the theoretical results.

**Audience:**

Yes

**Broader Impact Concerns:**

None, understanding an existing work.

**Claims And Evidence:**

Yes

**Requested Changes:**

- please check your baselines against SotA and/or add enough details of the experimental setup to explain the discrepancy in scores => critical
- re large tables => it'd be good to mark the 2nd best with a star => nice to have
- re high variance=> please perform a suitable statistical significance test (e.g., student t with bonferonni correction, or any other you can justify) to ensure that the improvments are statistically significant => critical
- re diffusion: please add more context/motivation for the reader => nice to have

**Strengths And Weaknesses:**

Strengths:

+ the papers exposition is decently clear for such a specific technical topic with some exceptions noted below
+ a wide set of baselines in the empirical evaluation
+ while I did not have the chance to check the proofs in detail and am not an expert in framelets, I could not find any errors in the derivations in the time I could spend on it
+ the theoretical results are a nice addition to the strong empirical performance


Weaknesses
- some variables are not clearly defined (in particular, $Y$ an $\theta$, the latter might be somewhere in the paper, for the former I had to check Shao 2022, page 13
- empirical validation scores of some models do not align with https://paperswithcode.com/sota/node-property-prediction-on-ogbn-arxiv and other Sota (e.g.  GCN rank 68, 71%) and I did not find sufficient details on hyperparameters, parameter count or code used to explain this
- lots of comparisons are good, but large tables make it difficult to compare => see below
- pL-UFG and the prescribed LD/HFD methdos seem to suffer high variance => see below
- equivalence to a nonlinear diffusion is proven, but not conextualized. How should one intrepret this, why should we care?

---

> ### Author Response · Authors · 2023-09-07
> **Thanks for your valuable suggestions**
>
> Dear Reviewer:
>
> Thank you very much for considering our manuscript and providing insightful suggestions. We have carefully reviewed your comments and will revise our manuscript accordingly very soon.
>
> Best, Authors

---

> ### Author Response · Authors · 2023-09-19
> **Detailed Response to Reviewer me6p**
>
> Dear Reviewer:
>
> Thank you so much for your invaluable comments on our manuscript. Below, please check the detailed response to your questions/comments. We note that all changes are highlighted in blue in our revised manuscript.
>
> (1):  Reply to the question: some variables are not clearly defined (in particular  $\mathbf Y$ and $\theta$) , the latter might be somewhere in the paper, for the former I had to check Shao 2022, page 13.
>
> Response: Thank you very much for pointing out this. We have explicitly denoted the form of $\mathbf Y$ in equation (18) of page 7 and $\theta$ on page 3 (Table 1) in our revised manuscript.
>
> (2): Reply to the question:  Empirical validation scores of some models do not align with https://paperswithcode.com/sota/node-property-prediction-on-ogbn-arxiv and other Sota (e.g. GCN rank 68, 71\%) and I did not find sufficient details on hyperparameters, parameter count or code used to explain this.
>
> Response: Thank you for the question. In our original submission, we used the data split with 20\%, 10\%, and 70\% for training, validation, and test dataset which may cause the results to be different from the benchmarks (i.e., ogbn-arxiv). In our revised experiment,  we conduct the experiment under the public split (i.e., split arxiv data based on the paper years) and the updated results in our revised manuscript (Table 3 at page 24).
>
> (3): Rely to the suggestion: it'd be good to mark the 2nd best with a star.
>
> Response: Thanks again for your suggestion, followed by your suggestions, in our revised manuscript, the top two accuracies are highlighted in bold red and blue. Please check Table 3 and Table 4 on page 24 and 25. Furthermore, we also adjusted the size of tables to make them more clear.
>
> (4): Reply to the question on the relatively high variance of the learning results: please perform a suitable statistical significance test (e.g., student t with Bonferonni correction, or any other you can justify) to ensure that the improvements are statistically significant.
>
> Response: Thanks for the insightful suggestion. We first show the intuitive understanding on the potential reason of relatively high variance of our learning results. First, we recall that our proposed models, named as pL-UFG-LFD and pL-UFG-HFD, respectively, are with the propagation of graph framelet with controlled dynamic takes the form as:
>
> $\mathbf F^{(k + 1)} = \sigma\left(\mathcal W_{0,1}^\top {\rm diag}(\mathbf \theta_{0,1}) \mathcal W_{0,1} \mathbf F^{(k)} \widehat{\mathbf W}^{(k)} + \mathcal W_{1,1}^\top {\rm diag}(\mathbf \theta_{1,1}) \mathcal W_{1,1} \mathbf F^{(k)} \widehat{\mathbf W}^{(k)}\right),$
>
> after which the output node features will be propagated through $p$-Laplacian based implicit layer. See Section 3.4 of our revised manuscript for more details. In our experiment setting, we let $\text{diag}(\theta_{0,1}) = \mathbf I_N$ and adjust the quantity of $\theta_{1,1} = \theta \theta_{0,1}$, suggesting two fixed constants applied on the $\theta$(s) that previously learnable in graph framelet (i.e., equation 9 and 10). This reduction in the parameter space has potential in inducing a large model bias, i.e., missing the desired solution as it did previously. We have included the corresponding discussion on Section 4.4, page 23 of our revised manuscript.
>
> In terms of empirically verifying the significance of the model improvement, we note that although our learning results are with relatively high variance (standard derivation), the quantities of the standard derivation are still comparable to many included baseline results for example Computers for homophily graph and Wisconsin for heterophily graph. Here let us consider the experiment on Citeseer as an example. The average accuracy of 10 runs of the model pL-UFG2$^{1.5}$ is $76.12\pm0.82$, which is the second best result. } The 95\% confidence interval (CI) via the t-distribution is computed as $76.12\pm t_{0.025, 9}\times 0.82/\sqrt{10}$, yielding the CI [75.53, 76.70]. However, our proposed model (pL-UFG-LFD) is with the average learning accuracy of 77.39 which is outside the confidence interval, suggesting a significant improvement from our model. One can also observe that significant improvement of our model in the results of heterophily graphs.
>
> Finally, followed by your instruction, in our revised manuscript, we marked our model's learning accuracy with a star (*) if its improvement is significantly higher than the highest result our baselines, i.e., the learn accuracy is outside the 95\% confidence interval. Please check Section 4.3 on page 22 in the revised manuscript.
>
> (5): Reply to the question: Equivalence to a nonlinear diffusion is proven but how should one interpret this?
>
> Response: Thank you very much for your notification. Based on your comments, we have enriched our Remark 9 in our revised manuscript, which includes the interpretation of the equivalence relation at page 19. Please let us know if you have any other requirements.

---

> > ### Comment · Reviewer_me6p · 2023-09-19
> >
> > Thank your for the incorporation of my feedback and the responses to me and the other authors. I will need to digest these in time (modulo conference deadlines...) but already one minor suggestion for the final version: I suggest **bolding** the top accuracy and underlining the 2nd highest to help colorblind readers.

---

> > > ### Author Response · Authors · 2023-09-19
> > >
> > > Dear Reviewer:
> > >
> > > Thank you very much for your prompt response, we will revise our manuscript very soon based on your suggestion. Please let us know if any additional changes are required, but do take your time on preparing your conference.
> > >
> > > Best, 1375 Authors.

---

### Review · Reviewer_S7am · 2023-09-09

**Summary Of Contributions:**

The submission does an analysis of the graph p-Laplacian regularized framelet network. It focuses on three issues: whether the filtering operation can converge and whether the Dirichlet energy will not converge to zero. Then the submission proposes a method to control the frequency of filters in the model. The experiment verifies that the model with controlled frequency can adapt to different problems (homogeneous or heterogeneous graphs).

**Audience:**

No

**Broader Impact Concerns:**

No concerns.

**Claims And Evidence:**

No

**Requested Changes:**

I think this submission should be compressed and better organized.

**Strengths And Weaknesses:**

The submission is very hard to read, so I cannot verify all derivation details. I feel that the submission just stacks all calculations together. I don't know how these calculations are organized together.

1. Can you make it more concise? For example, are equations (3) -- (5) necessary? Why equation (2) is necessary? Which properties are essential for propagations in (9) and (10).

2 Can you improve the logical flow? The writing only lists the computation of different matrices, but why do we need to go from one step to the next one?  For example, why can we use approximations in (6) -- (8)?  Why do we want to have the regularization problem in (18)? How is the equation (19) related to (9) and (10)? What is the matrix $\mathbf{M}$ in (21)? Why do we need equation (20)?

3. Technical questions.
1) If $\theta_{r,\ell} \in mathbb{R}^N$ is learnable, then it grows with the size of the input graph. How can you handle inductive learning?
2)  It seems that Theorem 1 is very similar to Theorem 2 in (Fu 2022). What is the relationship?
3) In the analysis of energy behavior, is the propagation in (31) the same as previous propgations?
4) What is Dirichlet energy?
5) I am wondering whether we will need complex techniques to show Lemma 3. When $\mu$ is large, then the model naturally will learn more signals in Y.
6) Corollary 1 and 2 are not rigorous.

4. In section 4.3, you have listed the range of different hyper-parameters. How do you tune your hyper-parameters? On which nodes do you decide these parameters?  What are other hyper-parameters (e.g. sizes of W matrices)? How do you tune these hyper-parameters?

---

> ### Author Response · Authors · 2023-09-19
> **Detailed Response to Reviewer S7am Part 1**
>
> Dear Reviewer:
>
> Thank you so much for your insightful questions and suggestions. Please find below the detailed response for each of your comments/questions. Again, all changes are highlighted in blue in our revised manuscript.
>
> (1): Reply to the comment : The submission is very hard to read, so I cannot verify all derivation details. I feel that the submission just stacks all calculations together. I don't know how these calculations are organized together.
>
> Response: Thank you for your feedback. We've made significant improvements in the revised manuscript to enhance the organization of calculations and provide clearer explanations of our theoretical results and their connections. We have also strengthened the proofs. We aim to improve readability and hope these changes meet your expectations.
>
> (2): Reply to the question:  Can you make it more concise? For example, are equations (3) -- (5) necessary? Why equation (2) is necessary? Which properties are essential for propagations in (9) and (10).
>
> Response: Thank you very much for your comments. Based on your suggestions, we have added several necessary interpretations (highlighted in blue) on the underlying logic between the equations and theories. For the relevance of equations (2), (3)-(5), and their importance in the context of propagations in (9) and (10), we have included detailed discussion on these equations and related theories at page 4 and 5 in our revised manuscript.
>
> (3): Reply to the question:  Can you improve the logical flow? The writing only lists the computation of different matrices, but why do we need to go from one step to the next one? For example, why can we use approximations in (6) -- (8)? Why do we want to have the regularization problem in (18)? How is the equation (19) related to (9) and (10)? What is the matrix $\mathbf M$ in (21)? Why do we need equation (20)?
>
> Response: We thank again for this valuable suggestion. Similar to our responses for the first comment, followed by your instructions, we have largely enriched the context (logic) between equations in our revised manuscript. Furthermore, in equation (21), the notion of $M_{i,j}$ stands for the elements in the matrix $\mathbf M$ defined in equation (19). The inclusion of equation (20) is with the following two purposes. First, the $\zeta_{i,j}^\phi(\mathbf F)$ makes the form of $\mathbf M$ defined in (19) easier to denote. Second, in Lemma (1) we show $|\zeta_{i,j}^\phi(\mathbf F)|$ is bounded and according to the result of Lemma (1), in Theorem (1) (the weak convergence theorem) we show the whole $\mathbf M$ is bounded at the bottom of page 10. Therefore, it is convenient and useful for denoting $\zeta(\mathbf F)$ in equation (20).
>
> (4): Reply to the question: For technical details, if $\theta_{r,\ell}$ is learnable, then it grows with the size of the input graph. How can you handle inductive learning?
>
> Response: Thank you for your comment, we note that the size of parameters $\mathrm{diag}(\theta)$ is the number of nodes. This is manageable as the complexity is $O(N)$.
>
> (5): Reply to the question:  It seems that Theorem 1 is very similar to Theorem 2 in (Fu 2022). What is the relationship?
>
> Response: Thank you very much for this notification. We have noticed that we have not provide any justification on the similarity/difference between our conclusion and Theorem 2 in (Fu, 2022). Taking this opportunity, based on your suggestion, we have included some contents on illustrating the difference between two conclusions and the importance of deriving the convergence property of the implicit layer. Please kindly check the highlighted contents at the bottom of page 8 in our revised manuscript and let us know if any further clarification is required.
>
> (6):  Reply to the question: In the analysis of energy behaviour, is the propagation in (31) the same as previous propagation?
>
> Response: Thank you very much for your comment. We note that the generalized propagation included in equation (31) is a general form of the feature propagation on the graph convolution. As it can be simplified to many existing GNNs. More importantly, the purpose of equation (31) is to induce the form of generalized Dirichlet energy in equation (32), which measures to total variation of the node features. Then we show the propagation of pL-UFG defined in equation (19) can be treated as a special layer-wised version of the generalized energy and followed by our subsequent analysis.
>
> (7): Reply to the question on What is Dirichlet energy?
>
> Response: Thanks for the reminder. In our original submission, We firstly mentioned Dirichlet energy on page 11, and explicitly defined the energy latter at equation (43) page 15. In our revised manuscript, we show the explicit form of Dirichlet energy on both pages 12 and 14 (equation 40) in case of any confusion.

---

> > ### Author Response · Authors · 2023-09-19
> > **Detailed Response to Reviewer S7am Part 2**
> >
> > (8): Reply to the question:  I am wondering whether we will need complex techniques to show Lemma 3. When $\mu$ is large, then the model naturally will learn more signals in $\mathbf Y$.
> >
> > Response: Thank you for your comments and we agree with your opinion that a larger quantity of $\mu$ indeed let the model learn more signals in $\mathbf Y$. However, our purpose in Lemma 3 and the subsequent analysis is to explicitly show how the total variation of the node feature changes under the changes of $\mu$ and $p$, especially when model is with L/HFD dynamics. Accordingly, Lemma (3) and Remark (7) (previously as Corollary (2) in the original manuscript) show that if the model is under H/LFD, by setting sufficient large/small of $\mu$, the model will induce a stronger H/LFD, thus producing more adaptive power on hetero/homophily graphs.
> >
> > (9): Reply to the comment:  Corollary 1 and 2 are not rigorous.
> >
> > Response: We apologize for the potential non-rigorous presentation. In our revised manuscript, we have enriched the proof of Corollary 1 and changed Corollary 2 to Remark 7. For your convenience, we include them below:
> >
> > For Corollary 1 proof:
> >
> > The proof can be done by combining Proposition 3 and Proposition 2, with the former illustrates that when model is HFD, there will be no over-smoothing problem, and the latter shows that even when the model is HFD, the Dirichlet energy of the node features will not converge to 0. Accordingly, pL-UFG is capable of escaping from over-smoothing issue.
> >
> > For Corollary 2 (Now as remark 7):
> >
> > Based on the condition described in Proposition 3, when framelet is LFD, with sufficient small of $\mu$ or larger of $p$, it is not hard to verify that according to equation 38, the p-Laplacian implicit propagation further shrink the Dirichlet energy of the node feature produced from framelet, and thus achieving a stronger LFD dynamic.
> >
> > (10): Reply to the question: In section 4.3, you have listed the range of different hyper-parameters. How do you tune your hyper-parameters? On which nodes do you decide these parameters? What are other hyper-parameters (e.g. sizes of W matrices)? How do you tune these hyper-parameters
> >
> > Response: Thank you very much for your comment. We tuned our hyper-parameters using grid search method. The settings of hyper-parameters are exactly same as [Shao et al 2022] in order to make a fair comparison. The proposed models are conducted by simply setting the filtering matrices to their desired constants (for the purpose of L/HFD). Followed by your instructions, we have enriched the related discussion by adding the parameter searching space from [Shao et al 2022] at page 20 of our revised manuscript.
> >
> > [Shao et al 2022]: Zhiqi Shao, Andi Han, Dai Shi, Andrey Vasnev, and Junbin Gao. Generalized Laplacian regularized framelet gcns. arXiv:2210.15092, 2022

---

> > > ### Comment · Reviewer_S7am · 2023-10-02
> > > **Thank you for your effort in improving the work**
> > >
> > > Thank you for your effort in improving the work. However, I still feel that the motivation is still not clear.
> > >
> > > Another example of unclear writing: "In this part, we provide several additional definitions to formulate the model (pL-UFG) that we are interested in analyzing." The introduction before this paragraph focused on framelet models. However, after this sentence, the topic changes to pL-UPG. The transition just says that "we are interested in analyzing". But why is it interesting/useful? What is the relationship between pL-UFG and framelets? P-Laplacian regularization probably is the most important concept of the work, but I still don't know what it is (the concept, not the technique definition) and why it is useful.
> > >
> > > Overall, I feel that the first 7 pages contain too many details but not a big map.

---

> > > > ### Author Response · Authors · 2023-10-03
> > > > **Response to Reviewer S7am**
> > > >
> > > > Dear Reviewer:
> > > >
> > > > Thank you very much for your follow-up comments. We apologize for the unclear logic between different topics. Below are our detailed responses.
> > > >
> > > > (1): First, we believe that, as a revisiting research for a theoretical analysis on pL-UFG,  it is important to include
> > > > the detailed formulation of pL-UFG. As pL-UFG is a model built on the top of the original graph framelet model (UFG), introducing UFG before pL-UFG is necessary for a self-explained paper.  To highlight the link between UFG and pL-UFG, based on your instructions, we added the following contents at the beginning of the paragraph of defining pL-UFG at page 5 of our revised manuscript, and for your convenience, we include it as follows.
> > > >
> > > > *As a generalized framelet model incorporating with the so-called $p$-Laplacian energy regularizer, the pL-UFG, as we are going to define it later, has shown great flexibility in terms of adapting different types of graphs (i.e., homophily and heterophily) by efficiently adjusting the penalty strength from the regularizer, resulted in superior learning performance across various benchmark datasets [shao et al 2022]. To thoroughly define pL-UFG, we start by defining the $p$-Laplace operator as follows.*
> > > >
> > > > Furthermore, in our revised manuscript, we have included the contents of motivating pL-UFG and the relationship between pL-UFG and original framelet model in the paragraph at the middle of page 6.
> > > >
> > > >
> > > > (2): In regarding to the concern about the concept of the p-Laplacian regularization, in our revised manuscript, based on your suggestion, we have changed our interpretation of equation (17) in the middle of page 7, and explicitly shown that $\mathcal{S}^{\phi}_p(\mathbf F)$ is the p-Laplacian based energy regularizer, and it provides different penalty strength (due to the form of $\phi$) in regularizing the node features propagating through GNNs in general. This regularization may impose a weaker impact over neigbours node features of graph edges. Intuitively, for example, in the case of $p=1$ and $\phi(\xi) = \xi$, the term $\||\mathbf{x}_i-\mathbf{x}_j\||_1$ is more robust than the term in the classic Dirichlet energy $\||\mathbf{x}_i-\mathbf{x}_j\||^2_2$. This leaves the opportunity for larger gap between $\mathbf{x}_i$ and $\mathbf{x}_j$ over some edges.  Thus the major motivation of this revisiting paper is to provide a theoretical analysis on the guarantee of such as an impact favoured for heterophilic graphs.
> > > >
> > > > (3): Lastly, followed by your instructions, we rewrote the first paragraph in page 8 by re-stating the questions that we aim to resolve in our theoretical analysis to enhance the logic of the paper. For your convenience, we include the paragraph here as follows.
> > > >
> > > > *Finally, in this paper, we call the the iterative algorithm defined in equation (19) for realizing the implicit layer of the problem defined in equation (18) as the **pL-UFG model**. Although remarkable performance have been observed from pL-UFG, there are still some key properties of the model that require to be explicitly presented. For example, the convergence of the iterative algorithm in equation (19) for the implicit layer in equation (18), and how the implicit layer changes and interacts with the energy dynamic of the original framelet, what is the relationship between the propagation within implicit layer and other propagations such as diffusion on node features. We will explicitly show our theoretical results in the coming sections.*
> > > >
> > > >
> > > >
> > > >
> > > >
> > > >
> > > >
> > > >
> > > >
> > > >
> > > >
> > > >
> > > >
> > > >
> > > >
> > > >
> > > >
> > > > [Shao et al 2022]: Zhiqi Shao, Andi Han, Dai Shi, Andrey Vasnev, and Junbin Gao. Generalized Laplacian regularized framelet gcns. arXiv:2210.15092, 2022

---

### Review · Reviewer_Pz1U · 2023-09-11

**Summary Of Contributions:**

In this paper, the authors analyze a model of implicit layers for GNNs based on p-Laplacian regularization and framelets, proposed in previous work. They show that a previously-proposed iterative scheme decreases the regularized cost function; that the Dirichlet energy for some range of parameter do not converge to 0, and, relatedly, that it is possible to increase the high-frequency content of the signal (ie increase the dirichlet energy). They propose two new models with controlled dynamics, make some relation between the problem and non-linear diffusion, and finish by some experiments on homophilic and heterophilic graphs.

**Audience:**

Yes

**Claims And Evidence:**

No

**Requested Changes:**

There are too many individual changes to list, but I would advise a significant clarification of the notations, the contributions, the conclusions from the theoretical results, and a more thorough description of the proposed new layers.

**Strengths And Weaknesses:**

Strength:
- a rather diverse study on an interesting topic
- interesting experimental results on heterophilic and homophilic graphs

Weaknesses:

There are some major problems with this paper.

- the paper is very messy and quite confusing. Notations are all over the place, several notations are sometimes used for the same object, notions are defined again and again and it is very difficult to know what is what. Why do we need so many notions of energies? Or different notations for the same notion? (eq 35)

- A lot of theoretical results are imported from other papers, it is difficult to know what is new, what is not. The conclusions from the theoretical results are not clear.

- a lot of material are copy-pasted from other papers, sometimes to the point of using their notations and not define them in this paper. Eg, the intro is almost exactly that of Shao et al 2022 (which, admittedly, seems to be still a preprint), including their notation $\mathbf{Y}$ which is the core of the definition of the GNN layer but not defined here! Theorem 1, although indicated as a contribution, is exactly Thm 2 from Fu et al 2022, proof included. And so on.

- The proofs are quite wonky, it is difficult to assess their correctness. For instance, in Theorem 2, the authors prove, after much computation, that parameters can be chosen to "increase or decrease" the Dirichlet energy. Unless I am wrong, they prove this for one step of the algorithm, with a choice of parameter that depend on the current iterate. Thus, it does not prove that there is a fixed parameter for which the result holds asymptotically (although proving that $p$-Laplacian regularization does not yield a constant signal seems simpler than the attempted proof). In Lemma 3, I do not see how the authors go from (47) to the conclusion. And so on.

- I do not get the new models in section 3.4. Is it possible to have a more thorough description and maybe some pseudocode ?

- I am not sure what is proved in Lemma 4. What are the consequences of this result?

- The authors seem to imply that all $p$ are accessible, which is somewhat wrong. In the original paper Fu et al 2022, it is well described that only $p>1$ is admissible, to preserve convexity and smoothness of the optimization problem. Otherwise, the iterations do not converge to the solution of the problem. (the case $p=1$ is still convex, but requires non-smooth optimization techniques, see the literature on sparse optimization). Note that Theorem 1 in Shao et al 2022 is based on first-order conditions only, it is a priori only valid for $p>1$ (their paper is not published)

- the cited literature is quite recent, but the $p$-Laplacian problem itself (outside of GNN) is quite old, maybe more references can be given

Minor comment
- Remark 1: "graph highly heterophily" (or homophily)
- several definition of $p$-norm (what is "element-wise?"), not sure if consistent across the paper
- sometimes $p$ is defined outside of $\phi$, sometimes it refers to $\phi$
- prop 1: $k$ does not appear in the definitions
- Thm 1: "week convergence"
- thm 1: wrong definition of $\mathcal{L}$
- page 10: the fonction [...] and thus Lipschitz
- eq 44: same notation left and right
- Cora and Wisconsin are not synthetic datasets but real-world data

---

> ### Author Response · Authors · 2023-09-19
> **Detailed Response to Reviewer Pz1U Part 1**
>
> Dear Reviewer:
>
> Thank you so much for considering our manuscript and providing precious comments, suggestions and questions. We have carefully considered all your comments, and below please see our detailed response to each of them. We also note that all changes are highlighted in blue in our revised manuscript.
>
> (1): Reply to the question: The paper is very messy and quite confusing. Notations are all over the place, several notations are sometimes used for the same object, notions are defined again and again and it is very difficult to know what is what. Why do we need so many notions of energies? Or different notations for the same notion? (eq 35)?
>
> Response: Thank you for your comments. We apologize for any confusion on the notations. We have thoroughly check the notations by deleting unnecessary contents and adding more descriptions on the meanings of the notation. Furthermore, followed by your instruction, we enriched the notation table (Table 1) in our revised manuscript. In addition, as we noticed that there are numbers of necessary notations required to conduct our analysis, in our original manuscript, we stated that other than Table 1, we will re-mention the meaning of each notation when we firstly denoted them in case of any confusion. We believe that is the main reason for causing the confusions. Accordingly, we have largely revised the paper particularly on this problem. Hopefully the problem is significantly resolved to your and readers satisfaction.
>
> For the energy notation issues, based on your comments, in our revised manuscript, we have explicitly defined all types of energies for the clarification. Specifically, we provide Definition 7 to explicitly show the layer-wise Dirichlet energy of the node features induced from p-Laplacian based implicit layer (equation (19) at page 12 of our revised manuscript. We then include the reason of defining such layer-wised energy after Definition 7 by stating that such energy is dynamic along the iteration of the implicit layer (page 12), and our subsequent analysis will be focusing on this type of energy (i.e. Proposition 2). In addition, we have largely revised the proof of Proposition 2 in our revised manuscript, we hope our efforts can make the paper meet the reviewer's satisfaction.
>
> After proving the energy defined in Definition 7 is away from 0 at any iterative layer k and increase with the increase of $\mu$ and $p$ in Proposition 2, how the inclusion of the implicit layer affects the original energy dynamic of graph framelet becomes the next question. Accordingly, one shall measure the energy dynamic of framelet according to its feature propagation (i.e., equation 9, page 5 of revised manuscript), yielding the generalized framelet energy defined in equation 41, page 15. It is worth noting that as the energy defined for the p-Laplacian based implicit layer (Definition 7) is layer-wised, it is natural to measure its impact on the layer-wise Dirichlet energy of graph framelet. That is the reason we include layer-wise framelet energy in equation 42 at page 16. Lastly, to explicitly show how the feature changes from framelet to the implicit layer in an end-to-end manner, we proposed total Dirichlet energy on equation 44 page 16 to solidly illustrate the energy of nodes after a whole process of framelet plus implicit layer.
>
> Finally, based on your comments, we have revised the corresponding contents in Section 3.2 and 3.3, page 12 to 17 in our revised manuscript. We thank you again for the detailed suggestions.

---

> > ### Author Response · Authors · 2023-09-19
> > **Detailed Response to Reviewer Pz1U Part 2**
> >
> > (2): Reply to the comment:  A lot of theoretical results are imported from other papers, it is difficult to know what is new, what is not. The conclusions from the theoretical results are not clear. A lot of material are copy-pasted from other papers, sometimes to the point of using their notations and not define them in this paper. Eg, the intro is almost exactly that of Shao et al 2022 (which, admittedly, seems to be still a preprint), including their notation $\mathbf Y$ which is the core of the definition of the GNN layer but not defined here! Theorem 1, although indicated as a contribution, is exactly Thm 2 from Fu et al 2022, proof included. And so on.
> >
> > Response: Thank you very much for your notification. First, as this work is a revisiting work of the existing p-Laplacian framelet model, to build a reader-friendly environment, it is preferable to maintain the notation used in (Shao, 2022). In regard to the missing of the notation of $\mathbf Y$, followed by your suggestions, we have include the interpretation of $\mathbf Y$ in both notation Table (Table 1) and the place when we firstly denote it (under equation (18)) in our revised manuscript.
> >
> > Furthermore, in terms of the difference between our Theorem 1 and Theorem 2 in Fu et al 2022, we would like to highlight (1): Our convergence conclusion is proved for the implicit layer defined in equation (19), which is different to the one defined in Fu et al 2022. In fact, the model defined in Fu et al 2022 can be considered as a special case where $\phi(\xi) = \xi^p$. (2): Our proof is conducted based on solid settings/assumptions (e.g., different form of the function $\phi$ in Lemma 1), in order to illustrate the convergence of our generalized algorithm compared to Fu et al 2022.
> >
> > Finally, for the concern about many copies of the contents from other papers in Introduction section, we first thank the reviewer for this notification and note that the only similar part between our manuscript and Shao et al 2022 in Introduction section is in the first two paragraphs which only give a general review on the current GNN research. Furthermore, after introducing the GNNs, the motivation, the contribution of the paper and all other parts of the Introduction are completely different compared to Shao et al 2022. Nonetheless, we followed your comments and revised our Introduction section accordingly in our revised manuscript. Please let us know if further changes are required.
> >
> > (3): Reply to the comment: The proofs are quite wonky, it is difficult to assess their correctness. For instance, in Theorem 2, the authors prove, after much computation, that parameters can be chosen to "increase or decrease" the Dirichlet energy. Unless I am wrong, they prove this for one step of the algorithm, with a choice of parameter that depend on the current iterate. Thus, it does not prove that there is a fixed parameter for which the result holds asymptotically (although proving that p-Laplacian regularization does not yield a constant signal seems simpler than the attempted proof). In Lemma 3, I do not see how the authors go from (47) to the conclusion. And so on.
> >
> > Response: Thank you very much for your comment. We appreciate your concern over the confused wording presented in the original proof. We have done our best to make it more logically structured. For example, we have represent Proposition 2 with more precise meaning by deleting unnecessary contents and with more solidly definitions for example: layer-wised generalized Dirichlet energy of the implicit layer. It is our belief that the newly proposed conclusion and proof is more logic and transparent for understanding. Please check the newly revised Section 3.2 from page 12 to 14 in our revised manuscript for more details.
> >
> > In terms of the definition on the layer-wised Dirichlet energy, we think we have provided the reasons of defining such energy in the response of your first comment. Please kindly check our revised manuscript (page 12 and 16)  and let us know if further changes are still required.
> >
> > In regarding to how equation (47) leads to the conclusion of Lemma 3. We first note that we have revised the contents of Lemma 3 and the its proof to make it more precise and logically structured.  By directly comparing the quantity of total energy (equation 44, previously as equation 47) with layer-wised framelet energy at layer $k+1$, we can see that the total Dirichlet energy is with a higher amount. This suggests the implicit layer (with sufficient large of $\mu$ or small of $p$) further amplifies the energy of the node feature, thus resulting in a stronger HFD dynamic from end to end. Please check the proof of Lemma (3) at page 16 for more details. We thank you again for this constructive suggestion.

---

> > > ### Author Response · Authors · 2023-09-19
> > > **Detailed Response to Reviewer Pz1U Part 3**
> > >
> > > (4): Reply to the comment: I do not get the new models in section 3.4. Is it possible to have a more thorough description and maybe some pseudocode?
> > >
> > > Response: Thank you very much for your comment, we have enriched the Section 3.4 by providing more explicit interpretation on the model propagation.
> > >
> > > (5): Reply to the comment:  I am not sure what is proved in Lemma 4. What are the consequences of this result?
> > >
> > > Response: Thanks for notifying the potential confusion in regarding to the result of Lemma 4. We note that the proof of Lemma 4 is to show that pL-UFG is aligned with existing graph diffusion framework with two source terms. The result from Lemma 4, together with our previous results of pL-UFG via energy behavior (Proposition 2), interaction of between framelet and implicit layer (Lemma 3) and the conclusion on over-smoothing (Corollary 1) can be unified and forms a well defined framework in analyzing pL-UFG. Accordingly, we have enriched Remark 9 of our revised manuscript.
> > >
> > > (6): Reply to the comment:  The authors seem to imply that all p
> > > are accessible, which is somewhat wrong. In the original paper Fu et al 2022, it is well described that only $p>1$
> > > is admissible, to preserve convexity and smoothness of the optimization problem. Otherwise, the iterations do not converge to the solution of the problem. (the case $p=0$ is still convex, but requires non-smooth optimization techniques, see the literature on sparse optimization). Note that Theorem 1 in Shao et al 2022 is based on first-order conditions only, it is a priori only valid for $p>1$
> > > (their paper is not published).
> > >
> > > Response: Thanks for your insightful notification. We agree with your opinion on the quantity of $p$ that can not be set with any value, and we believe that this is a point that requires a clarification in the paper as our results usually with the setting of sufficient large of $\mu$ or small of $p$. Accordingly, we have added one additional remark (Remark 4) in our revised manuscript which contains a discussion on the value of $p$ and the some clarification of our results. Hopefully, the concern about the value of $p$ can be resolved.
> > >
> > > (7): Reply to the comment:  The cited literature is quite recent, but the p-Laplacian problem itself (outside of GNN) is quite old, maybe more references can be given.
> > >
> > > Response: Thank you for your suggestions, in our revised manuscript, we formally defined p-Laplacian operator as Definition 3 on page 6 and included several citations that regarding to its existence, geometrical property and boundary conditions on so-called $p$-Laplacian equation.
> > >
> > > (8):  Reply to the minor changes of the paper.
> > >
> > > Response: Thank you so much for pointing out these issues. We have carefully considered your suggestions and revised our manuscript accordingly. We hope changes we made based on your invaluable comments make our manuscript more qualified for publication.

---

### Author Response · Authors · 2023-09-19
**Summary of the Changes in Revised Manuscript**

Dear Action Editor and Reviewers:

We thank to all reviewers for your invaluable comments on our manuscript. We have carefully considered all of your comments/suggestions and largely revised our manuscript accordingly. The following summary outlines the key modifications made between the two manuscript versions, with all alterations highlighted in blue within our revised document.

1. We solidify our manuscript by introducing additional formal definitions and refining our proofs.

2. Significant efforts have been dedicated to enhancing the readability of our manuscript. We now provide more comprehensive explanations bridging the concepts introduced in each definition or model and their corresponding theoretical implications.

3.  We have bolstered our empirical study by providing more information on hyper-parameter tuning, offering more comprehensive discussion of the experimental results and the limitations of our model.

4. We have rectified grammatical issues and augmented the citation references throughout the revised manuscript.

Thank you again and look forward to hearing from you soon.

Paper 1375 Authors.

---

### Decision · Action_Editor_q6Ho · 2023-10-30

**Recommendation:** Reject

**Comment:**

Primary concerns raised by reviewers for the original submission were the readability of the manuscript, and the validity and presentation of theoretical results. To address these concerns, the authors made significant changes to the manuscript. However, reviewers felt that the writing was still difficult to follow, and this change, including the revision of proofs, was too much to verify at once in the rebuttal period. Hence, the correctness of new proofs could not be well verified, and a full round of review seems to be required. Especially technical concerns (about the dependence of parameters on the iteration and the choice of p) raised by Reviewer Pz1U still remain at this moment. Since the contents of the manuscript are potentially useful, I recommend the resubmission with further improvement in the presentation of the theory.

**Audience:**

This paper could be of interest to those doing research on this specific subfield.

**Claims And Evidence:**

This paper presents a comprehensive analysis of the "Generalized p-Laplacian Regularized Framelet GCNs" (pL-UFG) model that combines Chebyshev polynomial approximation of the graph Laplacian with wavelet transforms and p-regularization. The authors demonstrate the convergence of the iterative procedure with an appropriate choice of regularization weight. They also show that the model's parameters can be adjusted to control the Dirichlet energy of node embeddings, preventing over-smoothing even in the asymptotic regime. Additionally, they propose a method to control the frequency of filters in the model, making it adaptive to homophilic and heterophilic data, and establish the connection between the regularization and non-linear diffusion process. Finally, the authors validate their theoretical findings through empirical experiments.

**Resubmission Of Major Revision:**

The authors may consider submitting a major revision at a later time.